



# An updated aerosol simulation in the Community Earth System Model (v2.1.3): dust and marine aerosol emissions and secondary organic aerosol formation

Yujuan Wang[1], Peng Zhang[1], Jie Li[2], Yaman Liu[3], Yanxu Zhang[1*], Jiawei Li[4*], Zhiwei Han[4,5]

[1] School of Atmospheric Sciences, Nanjing University, Nanjing, Nanjing, China
[2] Key Laboratory of Atmospheric Environment and Processes in the Boundary Layer over the Low- Latitude Plateau Region, Department of Atmospheric Sciences, Yunnan University, Kunming, China.
[3] Zhejiang Institute of Meteorological Sciences, Hangzhou, China
[4] Key Laboratory of Regional Climate-Environment for Temperate East Asia, Institute of Atmospheric Physics, Chinese
Academy of Sciences, Beijing, China
[5] University of Chinese Academy of Sciences, Beijing, China

*Correspondence to*: Yanxu Zhang (zhangyx@nju.edu.cn) and Jiawei Li (lijw@tea.ac.cn)

**Abstract.** Aerosols constitute important substance components of the Earth's atmosphere and have a profound influence on climate dynamics, radiative properties, and biogeochemical processes. Here we develop updated emission schemes for dust,

sea-salt, and marine primary organic aerosols (MPOA) and augment formation reactions for secondary organic aerosol (SOA) by introducing updated parameterizations within the Community Earth System Model (CESM; version 2.1.3). The modified scheme shifts the original hotspot-like dust emission to a more continuous distribution, improving the dust aerosol optical depth (DAOD) simulations at stations in North Africa and Central Asia. Also, it results in shorter dust residence time, necessary for enhancing concentration simulations downwind of dust source regions. Modifications in the sea-salt emission scheme

include an update to sea surface temperature (SST) modulation and the introduction of a relative-humidity-dependent correction factor for sea-salt particle size. The effect of SST is much more significant compared to that of relative humidity. We then extend to incorporate emissions of marine primary organic aerosols (MPOA) as externally mixed with sea-salt aerosols, coupled offline with ocean component Parallel Ocean Program (POP2). The influence of phytoplankton species on modeling MPOA emissions is profound, highlighting the significance of biological diversity in shaping aerosol emissions. In

addition to these emission scheme improvements, we also refine the chemical mechanisms in the model. The irreversible aqueous uptake of dicarbonyl compounds is added as a new pathway for the SOA formation in the model. These improvements enrich the capability of the CESM by using ESM's intricate linkage between different spheres of the Earth system, thereby enabling a more comprehensive description of natural aerosol emission and chemical processes and their impacts.





## 1 Introduction

Aerosols play a critical role in shaping Earth's energy budget and atmospheric properties (Dickerson et al., 1997). Intercomparison among AeroCom models indicate that the highest emissions among aerosol species are from sea-salt aerosols, with dust aerosols following as the second-largest contributor (Textor et al., 2006). In terms of aerosol mass burden, dust

aerosols are the most dominant, constituting approximately 75% of the total atmospheric aerosol burden (Kok et al., 2021b). These aerosols from natural sources stand out as key component due to their abundance and distribution throughout the Earth's atmosphere. While secondary organic aerosols (SOA) form through atmospheric chemical reactions rather than direct emissions, they constitute a major component of fine particulate matter, profoundly impacting human health and climate (Heal et al., 2012; Jimenez et al., 2009).

Dust aerosols, primarily originating from arid and semi-arid regions, are aerosols with size distributions that are highly variable over time and space (Mahowald et al., 2014; Tegen and Lacis, 1996). Emitted into the atmosphere through wind erosion processes, dust aerosol emissions display significant spatial and temporal variability spanning multiple orders of magnitude, and is sensitive to climate, land-cover and land-use change (Kok et al., 2021a; Mahowald et al., 2006). Transported across continents and oceans, dust aerosols play a critical role in various Earth system processes (Kok et al., 2023), including air

quality and population health (Mallone et al., 2011), cloud condensation nuclei (CCN) and ice nuclei (IN) formation (Koehler et al., 2009; Murray et al., 2012), radiation absorption and scattering (Kok et al., 2017), and nutrient deposition in oceans (Schroth et al., 2009). However, accurate observations to quantify dust emissions and their three-dimensional distribution remain challenging. Robust modeling approaches are therefore needed to simulate the global dust cycle, particularly regarding the initial emission processes, to enhance our understanding of their impacts on the Earth system.

Sea-salt aerosols, generated through the breaking of sea waves, constitute a substantial fraction of atmospheric aerosols in and around the oceans. Locally, sea-salt aerosols can affect the microphysical characteristics of ocean clouds (Platnick and Twomey, 1994), the intensity of tropical cyclones (Jiang et al., 2019) and even El Niño variability (Yang et al., 2016). Expanded to industrialized regions, investigations suggest the potential for sea-salt aerosols to moderate the direct radiative forcing of anthropogenic aerosols (Chen et al., 2020; Liao and Seinfeld, 2005). Yet, due to the scarcity of comprehensive

measurement of spatial and temporal evolution of sea-salt emissions, the emission estimation mainly relies on model simulations. As sea-salt aerosol emissions vary with ocean surface conditions (e.g., wind speed, seawater temperature and ambient humidity), it is necessary to incorporate these influencing factors in parameterization schemes (Lewis and Schwartz, 2004). Here, we seek to improve representation of dust and sea-salt aerosols through updates to foundational portrayal and modification terms in emission schemes.

While it is generally sea-salt aerosols that dominate the overall mass loading of sea spray aerosols, the biogenic organic fraction, known as marine primary organic aerosols (MPOA) has also been found to make up a significant portion of the



submicron aerosol mass concentrations during plankton bloom periods (O'Dowd et al., 2004). Given the complexity of refined parameterization of the organic content of the sea surface microlayer at the sea-air interface (Burrows et al., 2014), many recent MPOA emission schemes are built on sea surface chlorophyll-*a* concentration ([Chl *a*]) as an indicator for the organic mass

fraction of sea spray aerosols (Gantt et al., 2009; O'Dowd et al., 2008; Rinaldi et al., 2013). This correlates with the use of [Chl *a*] as a proxy of marine phytoplankton biomass (Cullen et al., 1982). Previous studies have utilized the global satellite-retrieved observation record of [Chl *a*] (Gantt et al., 2012; Meskhidze et al., 2011; Zhao et al., 2021). Here we exploit the strength of multi-sphere modeling in Earth system models to employ ocean biogeochemistry model in the simulation of MPOA emissions. The influence of different phytoplankton functional types could also be explored (Langmann et al., 2008; Roelofs,

2008; Spracklen et al., 2008).

Secondary organic aerosol, as a major component of the global submicron atmospheric organic aerosol, is formed by the oxidation of anthropogenic and biogenic volatile organic compounds (VOCs) (Kanakidou et al., 2005; Tsigaridis et al., 2014; Shrivastava et al., 2017). Its formation and subsequent dispersion affect air quality, climate forcing, and human health on a global scale (Hallquist et al., 2009). In addition, previous studies have shown that SOA plays a significant role in the regional

occurrence of severe haze pollution events (Huang et al., 2014; Li et al., 2021). Constrained by the complexity of the chemical composition and formation process of SOA, its representation in atmospheric chemistry models varies from the simplified approach using prescribed SOA emissions based on proportional values of precursor emissions (Chin et al., 2002; Colarco et al., 2010), to the advanced Volatility Basis Set (VBS) approach (Donahue et al., 2006; Hodzic et al., 2016). Due to the uncertainties and limited knowledge, current model simulations and SOA observations are still highly uncertain (Tsigaridis et

al., 2014). Li et al., (2021) found that the aqueous uptake of dicarbonyls is an important pathway for the formation of SOA, especially during haze events. Here, we include this new pathway in the chemical mechanism to investigate its impact on SOA formation.

This study focuses on improving the representation of atmospheric aerosols based on the conceptualization of CoAerM (the Common Aerosol Module), which is derived from previous work (Han et al., 2004; Li et al., 2021, 2024; Li and Han, 2016).

The modifications encompass natural aerosol emissions such as dust, sea-salt, and MPOA, as well as the formation of SOA, within the framework of the Community Earth System Model (CESM). CESM stands as a comprehensive framework comprising of sophisticated atmosphere, ocean, land, sea-ice, land-ice, runoff, and wave model components (Danabasoglu et al., 2020). This framework provides an expansive suite of options for configuring model components and physical parameterizations, enabling simulations of atmospheric composition changes and aerosol behaviors interacted with intricate

elements in Earth system. The updated aerosol schemes we develop is embedded into the atmospheric component of the CESM version 2.1.3., Community Atmosphere Model (version 6; CAM6). More specifically, we employ an integration of the Community Land Model (version 5; CLM5) and Parallel Ocean Program (POP2). The effects of the modified schemes and the sensitivity to specific changes are then described and compared. We organize the paper as follows: Sect. 2 presents the overall methodology, including detailed descriptions of the modifications to schemes and sources of measurements and satellite



retrievals. Section 3 evaluates simulated emissions and concentrations through comparison with observations and examines individual effects of scheme modifications. Summarized conclusions are provided in Sect. 4.

## 2 Methods and materials

### 2.1 Modifications to the schemes

#### 2.1.1 Dust emission scheme

**Vertical dust flux**

Online calculated dust emissions from wind erosion generally depend on the wind shear stress near the land surface and combine with characterizations of land surface properties, including vegetation cover, soil properties, and surface roughness. These parameterizations are either based on experiments (Gillette and Passi, 1988; Marticorena and Bergametti, 1995) or derived from microphysical processes (Shao, 2004). The standard dust emission scheme in CAM6 follows the Dust

Entrainment and Deposition (DEAD) model (Zender et al., 2003a), which is a semi-empirical expression. The total vertical dust flux ($F_{dust,j}$) emitted into size bin $j$ is modeled from the initial vertical emission flux calculated in CLM5 ($\varphi_{CLM,j}$) when the friction velocity ($u_{*s}$) exceeds the threshold friction velocity ($u_{*t}$):

$$F_{dust,j} = TS\varphi_{CLM,j} \tag{1}$$

$$\varphi_{CLM,j} = f_b \alpha c_s \frac{\rho_a}{g} u_{*s}^3 \left(1 - \frac{u_{*t}}{u_{*s}}\right)\left(1 + \frac{u_{*t}}{u_{*s}}\right)^2 \sum_i \chi_{i,j}, \quad u_{*s} > u_{*t} \tag{2}$$

Here $T$ is a global tuning factor; $S$ is an empirical geomorphic dust source function, also referred to as the soil erodibility factor; $f_b$ represents the bare soil fraction of grid cell suitable for dust entrainment; $\alpha$ is the sandblasting mass efficiency as a function of soil clay content; the saltation constant $c_s$ is set to 2.61; $\rho_a$ is the air density; $g$ is the gravitational acceleration; $\chi_{i,j}$ is the mass proportion from each source mode $i$ transported in the bin $j$. Note that friction velocity ($u_{*s}$) already accounts for the increase in friction velocity caused by the saltation process, as known as the Owen's effect (Owen, 1964). We modify the power law

relationship calculated in CLM5 based on (Gillette and Passi, 1988):

$$\varphi_{CLM,j}{}' = C_{GP} f_b{}' u_{*s}^4 \left(1 - \frac{u_{*t}}{u_{*s}}\right)\sum_i \chi_{i,j}, \quad u_{*s} > u_{*t} \tag{3}$$

where $C_{GP}$ is the calibration factor from Gillette–Passi scheme (set to $1.4 \times 10^{-15}$). It is noted that in all equations herein, the prime symbol ′ denotes the variables in the modified schemes.

**Surface roughness correction factor**

Threshold friction velocity ($u_{*t}$) represents the minimum velocity at which dust particles begin to move, and is commonly expressed as:

$$u_{*t} = u_{*t0}(D_{osp}) \times CF_w \times CF_d \tag{4}$$





where $u_{*t0}$ is the ideal threshold friction velocity, which depends on optimal saltation soil particles ($D_{osp}$); $CF_w$ and $CF_d$ are correction factors (CF) introduced to account for the soil moisture inhibition effect and the drag partition effect on $u_{*t}$,

respectively. In CAM6-DEAD, the soil moisture CF adopts the parameterization of Fécan et al., (1999), but the drag partition CF is inactive. We activate the drag partition parameterization (Darmenova et al., 2009; Marticorena and Bergametti, 1995; Tian et al., 2021) in this updated scheme to include the effect of non-erodible roughness elements. The $CF_d'$ is expressed as:

$$CF_d' = \left\{ 1 - \frac{\ln\left( z_0 / z_{0s} \right)}{\ln\left[ 0.7 \left( 12255 / z_{0s} \right)^{0.8} \right]} \right\}^{-1} \tag{5}$$

where $z_0$ and $z_{0s}$ are aerodynamic roughness length and smooth roughness length.

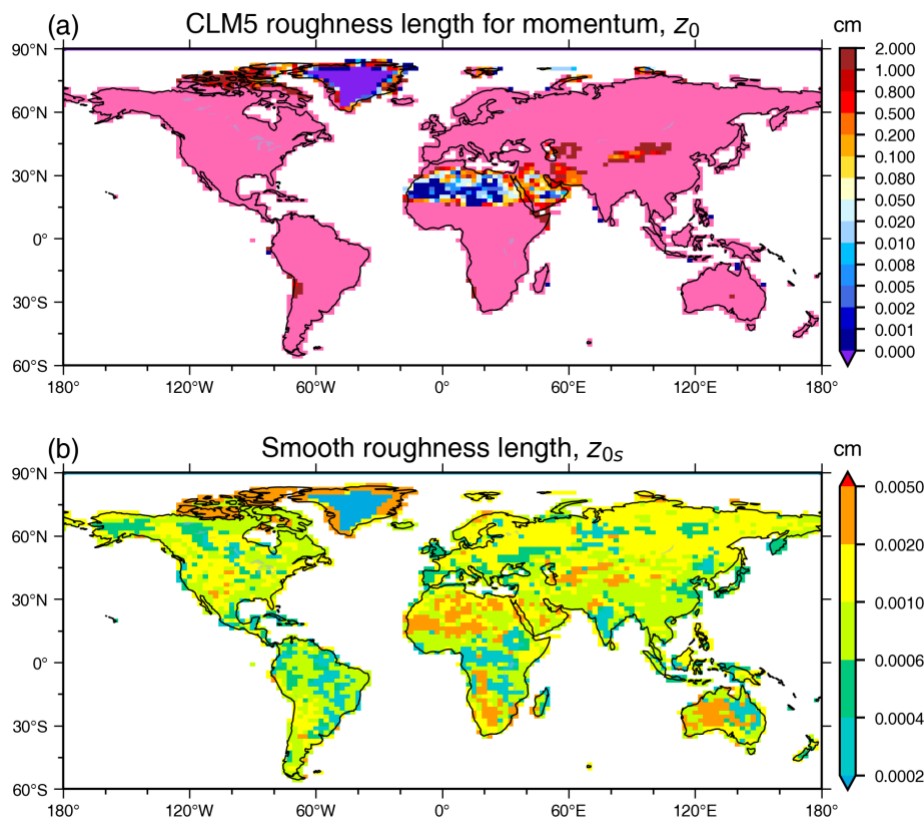

**Figure 1: Aerodynamic roughness length $z_0$ (a) and smooth roughness length $z_{0s}$ (b) used in updated dust emission schemes.**

The aerodynamic roughness length ($z_0$) refers to the roughness length of the exposed ground including the nonerodible elements, which dissipate part of the wind momentum for soil particle saltation. Zender et al., (2003a) used a globally constant





value of 0.01 cm for $z_0$. In this work, we choose to use the "patch roughness length over vegetation, momentum" computed

natively in CLM5. As shown in Fig.1a, the updated $z_0$ range is visibly extensive in comparison to the constant 0.01 cm.

The smooth roughness length ($z_{0s}$) is defined as the roughness length of the uncovered ground, that is, the roughness length

of the potentially erodible portion of the ground removed of any nonerodible elements. Zender et al., (2003a) used a globally

constant value of 0.0033 cm. This was estimated from a relation to the soil texture (Marticorena and Bergametti, 1995) when

assuming the area-mean diameter of particles to be 0.1 cm:

$$z_{0s} = \frac{D_{p,dust}}{30} \tag{6}$$


Here $D_{p,dust}$ is the underlying particle size. In this work, we introduce the geographic variability in $D_{p,dust}$. We first categorize

the soil texture in the model according to the USDA soil textural triangle (Soil Texture Calculator USDA, 2023) based on the

CLM-provided clay and sand content of the underlying soil. The result distribution is shown in Fig.S1. By corresponding to

the soil aggregate particle size distribution of different texture classification, $D_{p,dust}$ can be represented as the median diameter

of the coarse mode (Mokhtari et al., 2012; Tian et al., 2021). $z_{0s}$ can then be calculated from Eq.(6) (Fig.1b). Following the

application of the surface roughness correction factor, the simulated threshold friction velocity exhibits comparable changes,

with variations that differ across different regions (Fig.S2).

**Vegetation effects**

The original scheme employs $f_b$ to define the possible regions for wind erosion, which is expressed as:

$$f_b = (1-f_{lake})(1-f_{snow})f_{liq}(1-\frac{LAI}{LAI_t}) \tag{7}$$


where $f_{lake}$ and $f_{snow}$ are the lake and snow cover fraction of grid cell; $f_{liq}$ is the soil liquid water fraction in the top layer; LAI

is short for total leaf area index, which is used to indicate the inhibiting effect of vegetation cover on dust emissions. The

threshold LAI, $LAI_t$, is set to 0.3, above which dust emissions are assumed none (Mahowald et al., 2006). Yet experiments

shows that dust emissions could occur with high vegetation cover (Okin, 2008). In this work, we employ the reduction factors

(RF) of different vegetation cover to characterize the vegetation effects (Han et al., 2004; Park and In, 2003). Thus, the

modified expression of $f_b'$ is:

$$f_b' = (1-f_{lake})(1-f_{snow})f_{liq}\sum_{PFT}(1-f_i RF_i) \tag{8}$$

where PFT is short for plant functional type. $f_{pft}$ and $RF_{pft}$ are the fraction of grid cell and reduction factor of the $i$th type of

PFT prescribed from CLM5. To include all the effects of PFTs within a land unit, we do the sum at this subgrid scale. We

show the seasonal variability of the updated $f_b'$ in Fig.2. It is clear that the primary dust source regions around the globe

exhibit relatively high $f_b'$ values, indicating areas susceptible to wind erosion. Furthermore, regions with distinct seasonal





variations are concentrated in Central and East Asia, particularly during boreal winter (DJF), when the $f_b'$ reaches its lowest

values in these regions.

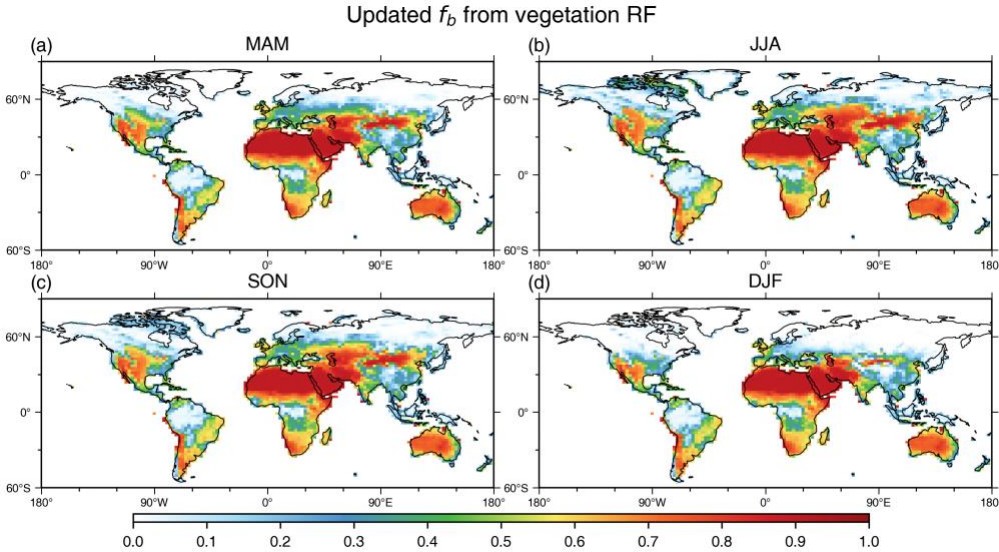

Updated $f_b$ from vegetation RF

(a) MAM (b) JJA

(c) SON (d) DJF

**Figure 2: Seasonal mean of updated $f_b'$ calculated based on vegetation reduction effects: (a) March-April-May (MAM), (b) June-July-August (JJA), (c) September-October-November (SON), and (d) December-January-February (DJF). The units are dimensionless.**

The original scheme also uses an empirical source function generated from geomorphic information of dust provenance to shape the distribution of dust emissions toward the so-called "preferential" regions (Albani et al., 2014; Mahowald et al., 2006; Zender et al., 2003b). However, the application of source function has been found to underpredict the dust emission in East Asia as dust source regions are not well characterized (Wu et al., 2021, 2019). Besides, the constraint of the source function on the modeling of dust emission suffers from the characteristics of the hotspot-like, discrete distribution (Fig.S3). Here we integrate land use data to modify the spreads in dust source region. Dust emissions are allowed in barren ground and areas with some shrubs or low grass.

**Mode mapping**

Subsequent to the computation of the total dust flux in CLM5, CAM5 allocates the emitted dust aerosols in the aerosol module. In this study, the four-mode version of Modal Aerosol Module (MAM4) (Liu et al., 2016) is used. Dust aerosols are distributed into three modes: an Aitken mode, an accumulation mode, and a coarse mode with emission diameter ranges of 0.01–0.1 μm, 0.1–1.0 μm, and 1.0–10.0 μm, respectively. The original prescribed emission mass proportion is derived from the "brittle fragmentation" theory (Kok, 2011), allocating fractions of 0.00165%, 1.1%, and 98.9% to the three modes, respectively. Here we modify the apportioning fractions to the accumulation mode and coarse mode according to the observed mass-size distribution of vertical dust flux in Chinese major source regions (Han et al., 2004) to 14% and 86%, respectively. The mass





fractions for the two modes are determined using the weighted mean distributions and mode diameter ranges (Li and Han, 2016).


### 2.1.2 Sea-salt emission scheme

Emissions of sea-salt aerosols are mostly modeled according to the mechanism of the bursting of whitecap bubbles entrained in breaking waves caused by surface wind. In the default emission scheme in CAM6, the quantification of emitted sea-salt particles with dry particle radius ($r_{dry}$) range of 1.4–5 μm is thus expressed as a source density function dependent on wind

speed drawn from laboratory observations (Monahan et al., 1986):

$$\frac{dF_{sea-salt,N}}{dr_{80}} = 1.373 U_{10}^{3.41} r_{80}^{-3}\left(1+0.057 r_{80}^{1.05}\right)\times 10^{1.19e^{-B^2}} \tag{9}$$

where $r_{80}$ is the sea-salt particle radius at 80 % ambient RH (~1.814 $r_{dry}$); $U_{10}$ is the 10-meter wind speed; $B=(0.380-\log r_{80})/0.650$. For sea-salt particles with $r_{dry}$ ranging from 0.01 to 1.4 μm, a scheme that considers the impact of SST on whitecap cover is used (Mårtensson et al., 2003). The source density flux is expressed as


$$\frac{dF_{sea-salt,N}}{d\log D_{dry}} = \left[a_k(D_{dry})T+b_k(D_{dry})\right]W \tag{10}$$

where the whitecap cover (fraction) is $W=3.84\times 10^{-4}U_{10}^{3.41}$; $D_{dry}$ is the dry particle diameter; $T$ is the SST (˚C). $\left[a_k(D_{dry})T+b_k(D_{dry})\right]$ is a polynomial term representing the SST effects with coefficients that varies with particle size. It was determined by laboratory experiments on synthetic seawater.

In this work, the sea-salt emission scheme is refined by incorporating two key modifications. First, the dependence of source

function on SST is updated using an empirical parameterization derived from cruise measurements (Jaeglé et al., 2011). We also employ an extended source function, which has been optimized for particles with radius under 0.1 μm (Gong, 2003). The modified source density function is as follows.

$$\frac{dF_{sea-salt,N}'}{dr_{80}} = (0.3+0.1\times T-0.0076\times T^2+0.0021\times T^3)\times$$
$$1.373 U_{10}^{3.41} r_{80}^{-A'}\left(1+0.057 r_{80}^{3.45}\right)\times 10^{1.607e^{-B'^2}} \tag{11}$$

Where $A'=4.7(1+30 r_{80})^{-0.017 r_{80}^{-1.44}}$ and $B'=(0.433-\log r_{80})/0.433$.

Second, a RH-dependent sea-salt particle size-correction factor is introduced, representing the influence of ambient humidity on sea-salt emission (Zhang et al., 2005, 2006). As previously described, sea-salt source functions are mostly presented in the form of $r_{80}$-based and $r_{80}$ is usually set as a multiple of $r_{dry}$. By incorporation the information of ambient RH, the source function can be expressed as



$$\frac{dF_{sea-salt,N}{}'}{dr_{dry}} = C^{80}\, \frac{dF_{sea-salt,N}{}'}{dr_{80}} \tag{12}$$

where $C^{80}$ is the correction term.

$$C^{80} \cong 1.82 \left( \frac{1-RH}{2.0-RH} \right)^{\frac{1}{3}} \tag{13}$$

### 2.1.3 Implementing MPOA emission into CESM

In the MAM4 module, the fourth aerosol mode is primary carbon mode used to investigate the ageing of primary carbonaceous

aerosols (Liu et al., 2016). Aerosol species of black carbon and primary organic matter (pom) are emitted into primary carbon

mode and then aged to the accumulation mode. The original emission setting, however, referred to primary organic aerosols

emitted to the primary carbon mode (denoted as "pom_a4") as terrestrial sources. These sources typically encompass terrestrial

biomass burning, fossil fuel, and biofuel combustion. In this study, we implement MPOA emissions into the accumulation

modes to be internally mixed with sea-salt. This configuration has been validated to be in best agreement with observations

among different assumptions of mixing state and amount changes (Burrows et al., 2018). We calculate MPOA emissions from

the organic mass fraction of sea spray aerosols ( $OM_{SSA}$ ) and updated sea-salt mass emissions ( $F_{sea-salt}{}'$ ):

$$F_{MPOA} = F_{sea-salt}{}' \times \frac{OM_{SSA}}{1-OM_{SSA}} \tag{14}$$

Parameterization of $OM_{SSA}$ is related to [Chl $a$] and 10-meter wind speed following Gantt et al., (2011). The formula is

expressed as

$$OM_{SSA} = \frac{1}{1+\exp\left(-2.63[Chl\ a]+0.18U_{10}\right)} \left( \frac{1}{1+0.03\exp(6.81D_{dry})} + 0.03 \right) \tag{15}$$

And majority of previous modeling research on MPOA used satellite-based observational [Chl $a$] concentration (Gantt et al.,

2012; Wang et al., 2020; Zhao et al., 2021; Li et al., 2024). In this study, model simulated [Chl a] will be used instead.

### 2.1.4 Offline coupling with ocean biogeochemical component

To integrate modeled [Chl $a$] results in the MPOA emission scheme, our modification involves coupling with the Marine

Biogeochemistry Library (MARBL) embedded in the ocean component POP2. MARBL functions as a prognostic ocean

biogeochemistry model, facilitating the simulation of marine ecosystem dynamics and cycling of essential elements, including

carbon and nitrogen (Long et al., 2021). It offers the capacity for adaptable ecosystem configurations of varying complexity

by allowing modifications to phytoplankton and zooplankton functional types. In this study, we preserve the ecosystem

configuration in MARBL-CESM2, which explicitly includes three phytoplankton functional groups (diatoms, diazotrophs,

"small" pico/nano phytoplankton), and one zooplankton group.





Considering the computational efficiency, we use the offline approach to drive the MPOA emission calculation with surface [Chl $a$] of the three phytoplankton functional groups derived from pre-processed POP2 run. We set up an ocean biogeochemistry run with the standard ocean component set of CESM2 (G component set). This setup includes active POP2 and the sea-ice component, together with a data atmosphere and stub land component. The simulation is conducted at

approximately 1° resolution from year 2000 to year 2012 with the first ten years as spin-up time and the last 3 years for regridding. [Chl $a$] data are read directly during run time in CAM6-chem from netCDF-format files with a monthly interval.

### 2.1.5 SOA formation via aqueous reaction

The default SOA parameterization in CAM6-Chem uses the VBS scheme, which groups the precursors of SOA into five bins by volatility, with saturation concentrations ranging from 0.01 to 100 μg/m$^3$ at 300 K (Tilmes et al., 2019). CAM6-Chem

explicitly simulates nine types of SOA precursors that oxidized mainly by OH, O$_3$, and NO$_3$ to form gas-phase semivolatile condensable sources of SOA (see Table 1 in Tilmes et al., 2019). However, the irreversible aqueous uptake of dicarbonyls (mainly glyoxal and methylglyoxal) has not been integrated into the chemical mechanism, and this process is considered to contribute significantly to the formation and the total burden of SOA (Fu et al., 2008, 2009; McNeill et al., 2012; Li et al., 2021). We add the irreversible uptake of dicarbonyls gases (glyoxal and methylglyoxal) by aqueous particles in the model.

The uptake rate $k$ is given by

$$k = \left( \frac{r}{D_g} + \frac{4}{\nu\gamma} \right)^{-1} \cdot A \tag{16}$$

Where $r$ is the particle radius, $D_g$ is the gas-phase molecular diffusion coefficient, $\nu$ is the mean gas molecular speed, $\gamma$ is the reactive uptake coefficient, and $A$ is the aerosol surface area. The reactive uptake coefficient for glyoxal and methylglyoxal adopts 2.9×10$^{-3}$ following Li et al., (2021).


### 2.2 Model configurations and experiments

We run CESM2 with the following configurations. To reproduce more realistic meteorological conditions, we use the FCSD component set which couples CAM6-chem with active land and sea-ice components, as well as the data ocean and slab land ice components in all simulations. CAM6-chem is configured to run with the finite volume dynamical core and

troposphere/stratosphere chemistry. Anthropogenic emissions of other aerosols and precursors are from Climate Model Intercomparison Program (CMIP6) historical inventory (Eyring et al., 2016). The meteorological nudging applies to air temperature, relative humidity, and horizontal wind components using Modern-Era Retrospective Analysis for Research and Applications version 2 atmospheric forcing dataset (MERRA2, 2018) with a 6-hour relaxation. The accuracy of the modeled wind speeds is decisive for all aerosol emissions involved in this study. Sea surface temperature (SST) and sea-ice cover fields

are prescribed from historical data.



We consider a "CYCLE" simulation set, simulating from year 2009 to year 2012, with the first year as the spin-up period and the last 3 years used for analysis. CAM5 and CLM5 are executed at a horizontal resolution of 1.9° × 2.5° in latitude and longitude, and a vertical resolution of 32 levels. The simulation set consists of a control run that implements the default emission schemes, and another run incorporating all the aforementioned modifications. The results are analyzed with a focus

on the global scale, aiming to capture the stable state rather than abrupt perturbations.

A case study of a dust events in East Asia during March 2021 is conducted to evaluate the updated dust emission schemes. This simulation set employs a finer grid with a resolution of 0.9° latitude × 1.25° longitude since the analysis is focused on the regional scale. In this case, the simulation period spans from 1 January 2021 to 1 April 2021 with a spin-up time of more than two months.

A set of sensitivity experiments is prompted to evaluate the dependence of the sea-salt aerosol emission schemes on two major modifications. First, we test the impact of SST correction by comparing the original scheme with simulations using the source function proposed by Gong (2003, referred to as the *Gong* function hereafter) without SST correction and with the SST correction from Jaeglé et al. (2011). Second, we assess the influence of a RH correction by running simulations with and without the RH-dependent correction factor using the *Gong* function.

Also, experiments on MPOA emission schemes involves a comparison of MPOA emissions simulated from input [Chl *a*] from several different species sources. The SOA experiment includes a control group (default chemical mechanism) and an experimental group (modified chemical mechanism) to investigate the effect of dicarbonyls uptake on SOA formation.

### 2.3 Observational measurements

**Dust aerosol optical depth (DAOD) based on satellite retrievals**

We use dust aerosol optical depth (DAOD) data on two time scales in this study for analysis: one is directly calculated from MODIS/Aqua gridded products, and the other is a climatological dataset of global DAOD derived from MODIS aerosol retrievals (Song et al., 2021).

To facilitate comparison with the model results in the dust event case, we apply the formula proposed by (Pu and Ginoux, 2018), combining Ångström exponent ($\alpha$) and single-scattering albedo ($\omega$) to calculate the daily DAOD from the MODIS/Aqua

Level-3 atmosphere daily joint product (MYD08_D3, Collection 6.1) . The MYD08_D3 product provides 1° × 1° grid values of atmospheric aerosol particle parameters (e.g., aerosol optical depth, abbreviated as AOD, $\alpha$, and $\omega$) retrieved using the Deep Blue aerosol algorithm.

Regarding the mean state of the dust cycle, we compare the model results with the globally aggregated monthly mean DAOD dataset. This climatological dataset is also derived from MODIS/Aqua satellite retrieval, with a spatial resolution of 1° and

temporal coverage from 2003 to 2019 (Song et al., 2021).

**Coarse-mode AOD of AERONET stations**

We also use the coarse-mode AOD (CAOD) from ground-based Aerosol Robotic Network (AERONET) measurements (Holben et al., 2001) at selected stations to evaluate the model performance against observations. Monitoring stations located



in major dust source regions with data availability of at least one full year during the "CYCLE" simulation periods are considered valid (see Fig.3 orange dots). To align the AERONET CAOD record with model results of DAOD, we perform interpolation to get CAOD at 550 nm using the Ångström exponent from the AERONET level-2 data (

$$CAOD_{550\,nm} = CAOD_{500\,nm}\left(\frac{550}{500}\right)^{-\alpha_{500\,nm}}$$

). Evaluation metrics include the root mean square error (RMSE) and Kendall's correlation coefficient (R).

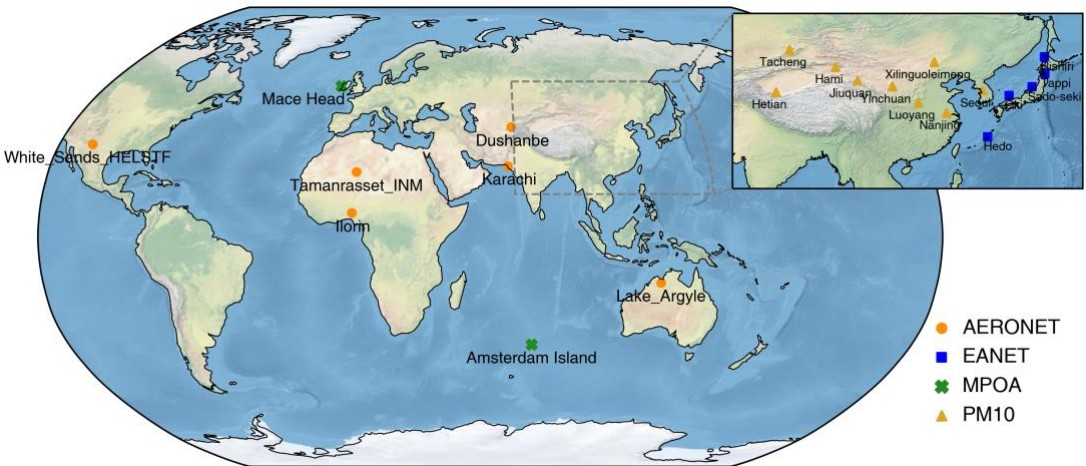

**Figure 3: Location of observations used in this study. Orange circles: AERONET sites. Blue squares: EANET stations. Green crosses: MPOA measurement site. Dark yellow triangles: ground-based PM$_{10}$ sites.**

**Aerosol concentration of EANET stations**

To validate the model performance of simulating sea-salt aerosol, *in situ* ground observations of aerosol concentration in remote coastal areas are collected from EANET (Acid Deposition Monitoring Network in East Asia EANET). Locations are shown in Fig.3. We use the data for the "CYCLE" simulation period. In addition to the recorded PM$_{10}$ concentration, sea-salt aerosol mass concentration is calculated from ion concentration as (Quinn and Bates, 2005):

$$\left[Sea-salt\right] = [Cl^-] + 1.47\times[Na^+] \tag{17}$$

**MPOA concentration from publications**

To validate MPOA concentration, we use measurements reported in previous publications, which commonly refer to the water-insoluble organic mass (WIOM) fraction of submicron marine aerosols as MPOA. This differentiation is made as the soluble organic fraction represents the more oxidized portion originating from secondary source (Rinaldi et al., 2010). Two measurement sites are considered in this paper (Fig.3). One is Mace Head (53.33˚ N, 9.90˚ W) with temporal coverage of January 2002 to June 2009 (Rinaldi et al., 2013). The other is Amsterdam Island (37.80˚ S, 77.57˚ E) with temporal coverage of January 2005–November 2007 (Sciare et al., 2009). We apply an organic-mass-to-organic-carbon ratio (OM/OC) of 1.4 to align the WIOM measurements with simulated MPOA concentration.





**PM10 concentration from stations**

To evaluate the model simulation of dust event in East Asia, ground measurements of $PM_{10}$ concentration are collected at stations in China and South Korea. $PM_{10}$ measurements for Chinese stations are obtained from China National Environmental Monitoring Centre and data for South Korean stations are obtained from Air Korea websites (Air Korea stations: https://www.airkorea.or.kr/web/sidoQualityCompare?itemCode=10007, last access: 17 March 2024). Daily $PM_{10}$ concentrations are used. Locations are shown in Fig.3.

## 3 Results and discussion

We discuss the revised dust aerosol emission scheme in Sect. 3.1, where we evaluate the modifications by comparing simulations with measurements, followed by a detailed case study of a significant dust event that occurred in March 2021 in East Asia. In Sect. 3.2, revised sea-salt emission scheme is evaluated and the impacts of each of the two key modifications for sea-salt aerosols are then investigated. The simulation of the MPOA is evaluated in Sect. 3.3, followed by a discussion of the impact of phytoplankton species on MPOA emissions. The update of SOA formation scheme is discussed in Sect. 3.4.

### 3.1 Dust emission scheme

### 3.1.1 Model evaluation

To align model results more closely with observations, a global tuning factor is often used in global dust simulations. Earlier studies (e.g., Klose et al., 2021; Li et al., 2022) opted to tune global dust emissions to achieve a simulated global annual mean DAOD of approximately 0.03. This is a global-scale constraint proposed from multiple satellite retrievals combined with modeling analysis (Ridley et al., 2016). The CAM6-chem namelist variable (*dust_emis_fact*) is implemented for our dust tuning. This tuning parameter is influenced by factors such as the dynamical scheme, chemistry scheme, and resolution settings (Li et al., 2022). In our study, we set this parameter to 0.57 for the original simulations and 0.29 for the updated simulations to meet the DAOD constraint mentioned above.

| Dust diameter (μm) | Original dust emission (Tg yr⁻¹) | Updated dust emission (Tg yr⁻¹) | Original dust wet deposition[a] (Tg yr⁻¹) | Updated dust wet deposition[a] (Tg yr⁻¹) |
|---|---|---|---|---|
| **0.01-0.1** | 0.039 | 0.043 | 0.004 (21%) | 0.006 (29%) |
| **0.1-1.0** | 26 | 368 | 20 (75%) | 294 (79%) |
| **1.0-10.0** | 2348 | 2260 | 1458 (62%) | 345 (15%) |
| **0.01-10.0** | 2374 | 2628 | 1478 (62%) | 639 (24%) |

**Table 1: Global dust emissions and wet depositions in "CYCLE" simulation sets. [a] The ratio of wet deposition to total deposition (dry and wet deposition) is listed in parentheses next to wet deposition.**



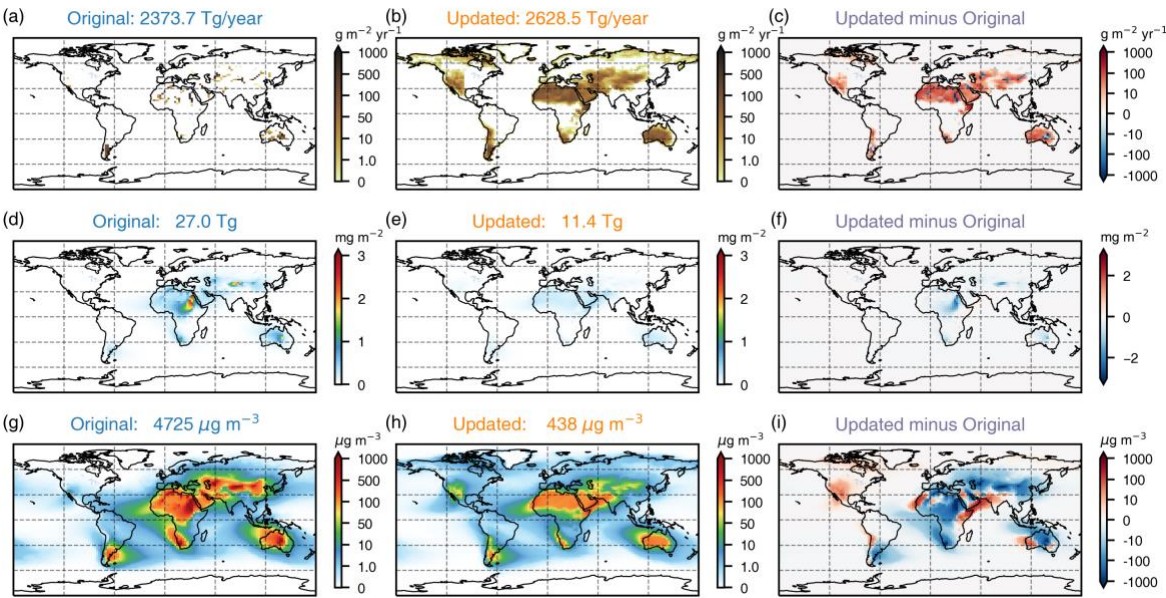

**Figure 4: Annual mean dust emissions (a, b, c), burdens (d, e, f), and surface dust aerosol concentrations (g, h, i) for the period from**
**2010 to 2012. Simulation results from the original (left) and updated (centre) emission schemes are shown, along with the differences**
**between the updated and original schemes (right). The global total dust emission, global total dust burden, or global maximum**
**surface dust concentration is also given at the top of each subplot.**

With the modifications applied, we simulate a global annual total dust emission of 2628 Tg, compared to 2374 Tg before
implementing these changes (Table 1). Both emission estimates align well within the estimated ranges reported by the
AeroCom models (514 to 4313 Tg: Huneeus et al., 2011; 1840 ± 902 Tg: Textor et al., 2006) and those from the CMIP5 models
that consider similar size ranges (735 to 8186 Tg), as summarized in Wu et al., (2020). The global annual dust deposition,
including both dry and wet depositions, is modeled as 2658 Tg in the updated scheme compared to 2390 Tg in the original
scheme. Wet deposition accounts for 62% of total deposition in the original simulation and is reduced to 24% in the updated
simulation (Table 1), consistent with the 12%-39% range of wet deposition fractions in CMIP5 models (Wu et al., 2020). The
adjustment of apportioning fractions for the accumulation and coarse modes, based on observed mass size distribution (Han et
al., 2004), results in an increase in annual dust emissions in the accumulation mode, reaching 368 Tg. The global distribution
of annual dust emissions, dust burdens and surface dust aerosol concentrations simulated with the default and modified
schemes is presented in Fig.4. Both the modified and the original schemes capture intense dust emissions from the major dust
emission regions globally, with the primary distinction lying in the spatial distribution. The original scheme targets dust
emissions to scattered 'hot-spots' based on the geomorphic erodibility factor (Fig. S3), while the updated scheme exhibits a
more continuous regional distribution of dust emissions. This is due to the adoption of land use distribution combined with

$f_b{'}$ to determine the areas where wind erosion processes are likely to occur (Fig. 3). In major dust-emitting regions, such as
North Africa, East Asia, Middle East, Central Asia, Australia, and South America, the updated scheme simulates broader dust





emission areas but with lower emissions than the original scheme. In the northwestern part of North America, the updated
scheme models larger dust emissions compared to the original scheme (7.6 g m$^{-2}$ yr$^{-1}$), with some areas experiencing
particularly strong dust emissions (~100 g m$^{-2}$ yr$^{-1}$). It is worth noting that the updated scheme simulates emissions in the high-
latitude dust (north of 50° N), which is absent in the original scheme. The simulated emissions, mostly below 10 g m$^{-2}$ yr$^{-1}$,
concentrate in the paraglacial area of the sub-Arctic, consistent with known dust observations and recorded dust storm
occurrences in this region (Bullard et al., 2016; Prospero et al., 2012).

Despite the updated scheme leading to a more extensive dust emission coverage, with emissions rising from 2374 Tg to 2628
Tg (an 11% increase), the global dust burden reduces notably from 27.0 Tg to 11.4 Tg (a 58 % decrease). This reduction is
likely attributed to a more uniform distribution of dust burdens, as the updated emissions are evenly distributed without
significant spikes in specific locations. For the distribution of near-surface dust aerosol mass concentrations, the updated
scheme simulates lower concentrations in the main dust source regions compared to the original scheme. Additionally, in the
downwind areas of certain major deserts, such as the Patagonian desert in South America, Nubian desert in North Africa, Gobi
desert in East Asia, the northern Middle East, and Australia, dust aerosols appear to have not spread to the same distances after
the modification, possibly influenced by the size distribution and deposition processes (Table 1). The overall dust aerosol
residence time (the ratio of burden to deposition) is shortened from 4.1 days to 1.6 days in the updated scheme, suggesting that
dust aerosols are not transported as far as previously thought. Nevertheless, in regions where emissions were augmented in the
updated scheme, such as the western Sahara, the Altiplano in South America (Lindau et al., 2021), and the southern part of the
Middle East (including the Horn of Africa), dust concentrations increase after the modification. Additionally, the increase in
dust emissions simulated by the updated scheme in southwestern North America leads to a substantial simulated dust aerosol
concentration, reaching a regional mean of 18 μg m$^{-3}$. This is close to the seasonal peak measured by a monitoring network in
the United States (Hand et al., 2017).

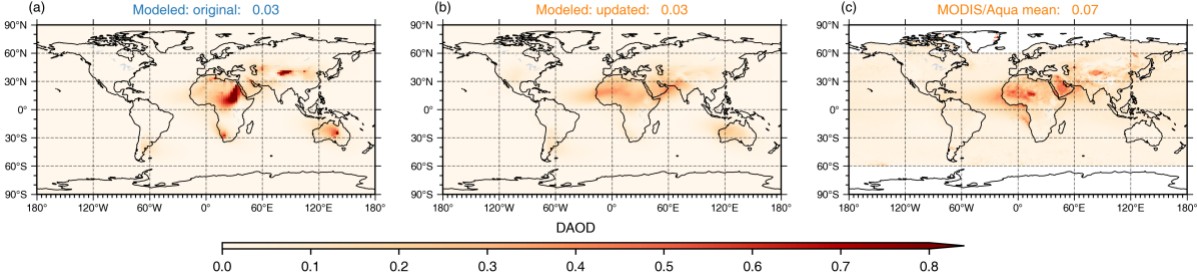


**Figure 5: Annual mean DAOD from the original (a), the updated (b) simulation results and MODIS/Aqua observation (c) (Song et
al., 2021) for the period from 2010 to 2012.**

In line with the distribution of dust burdens, the updated scheme simulates a smaller DAOD over major dust source regions
compared to the original scheme (Fig. 5a and 5b). Observations from the climatological dataset derived from MODIS/Aqua
show the maximum DAOD over the central and western parts of North Africa, the Middle East, and the Taklamakan in East
Asia (Fig. 5c). In comparison with the observation, the updated scheme captures the regional maxima of DAOD over central



and western North Africa and the Middle East, albeit underestimating it, in particular over central North Africa. The updated scheme also reproduces the DAOD distribution in Central and East Asia as the observations, but it tends to overestimate DAOD near Thar and underestimates DAOD near Taklamakan. The original scheme, on the other hand, simulates the regional

extremes of DAOD in central Australia and South Africa, which are not as evident in the observations. Overall, the updated scheme provides simulation results for DAOD that are closer to observations than the original scheme.

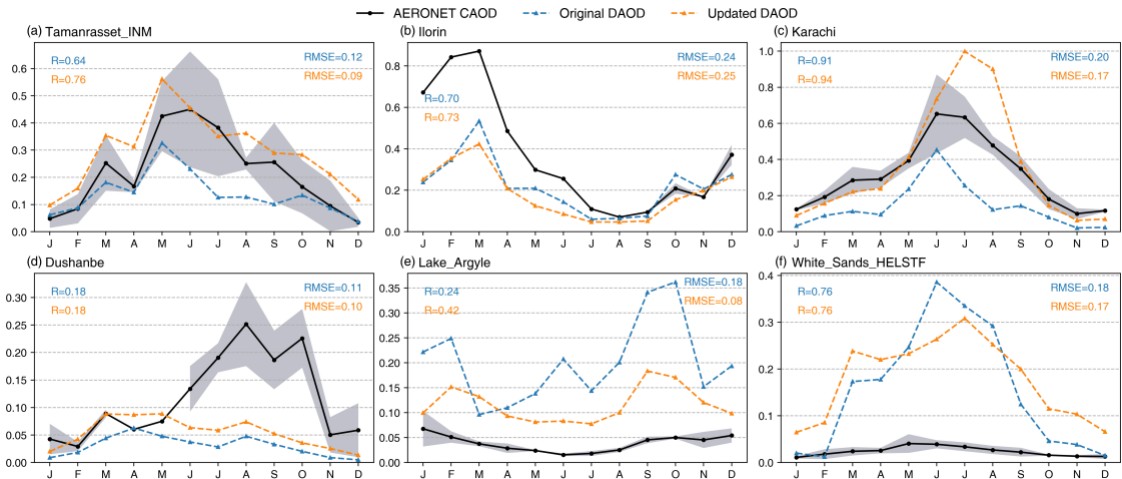

**Figure 6: Seasonal cycle (a-f) of monthly mean simulated DAOD (coloured lines) and AERONET CAOD (black lines) at selected stations (see Fig.3). The blue and yellow lines represent the simulations from the original and updated schemes, respectively. The**
**RMSE and R are noted in corresponding colored font for each simulation. The shading on the observations illustrates the standard deviation of the monthly mean CAOD over the months with sufficient data. North Africa: (a-b), Middle Asia: (c-d), Australia: (e), and North America: (f).**

We evaluate the month-to-month variability of the modelled DAOD using the CAOD obtained from AERONET measurements. The AERONET CAOD in dusty regions can be used for comparison with DAOD (Pu and Ginoux, 2018).
Presented results are of the stations that locate in the main dust source regions (North Africa, Middle East and Middle Asia, Australia, and North America) (Fig.3).

At two stations located close to the dust source: Tamanrasset_INM (North Africa) (Fig. 3a), Karachi (Central Asia) (Fig. 3c), the original scheme underestimates DAOD compared to the observations but captures the seasonal variability within the year. The metrics of the DAOD time series indicate that the updated simulations exhibit better correlation and smaller RMSEs at
these two stations. Especially for the Karachi station, despite the overestimation of the DAOD peak during the summer months of JJA, the updated scheme simulates DAOD that align notably closer with observations. For sites in downwind regions with some distance from the dust source: Ilorin (North Africa), Dushanbe (Central Asia) and White_Sands_HELSTF (North America), both the original and updated scheme results are in better agreement with AERONET for months of the year with less intense dust, but deviate considerably for dust peak months. Finally, at stations where the results of the original schemes
differ significantly from observations, such as Lake_Argyle (Australia), updated scheme shows improved correlation results





but still simulates poorly. This discrepancy may be attributed to the coarse resolution of the model, which fails to simulate dust transportation.

### 3.1.2 Case: East Asia dust events in 2021 March

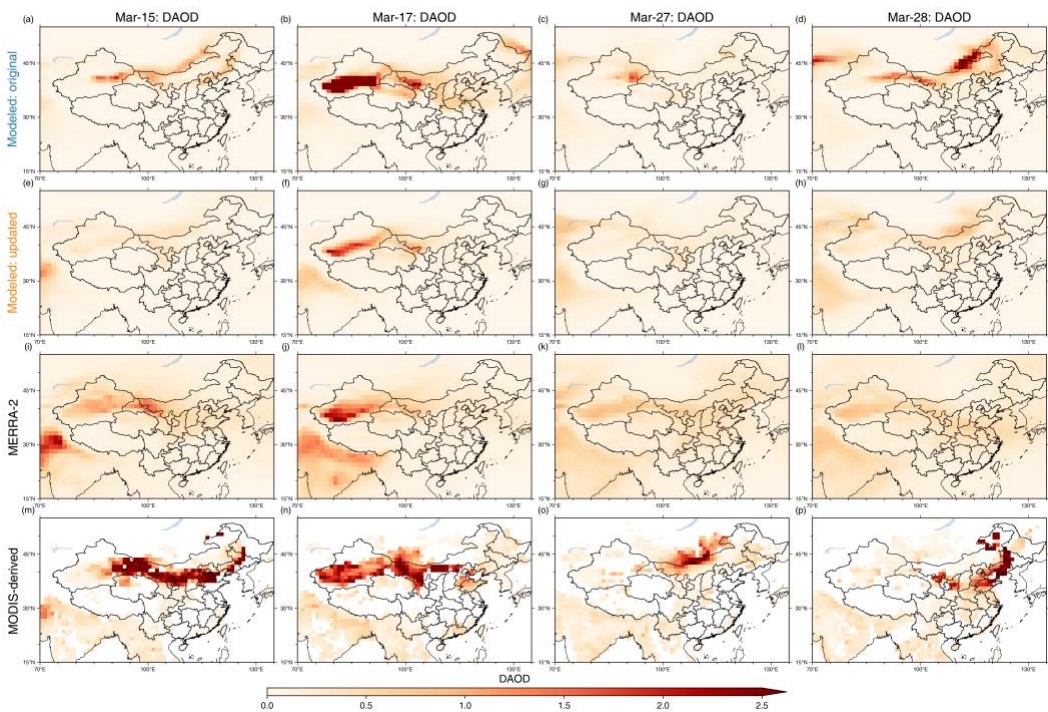

**Figure 7: Daily mean DAOD in East Asia during the 2021 March dust event. Simulation results from the original (the first row) and updated (the second row) emission schemes are shown, along with the MERRA-2 reanalysis (the third row) and MODIS derived results (the last row). Note that the MODIS DAOD are calculated according to Sect.2.3.**

On 15-20 March and 27-29 March, 2021, East Asia was hit by two intense dust events (WMO news report: https://wmo.int/media/news/severe-sand-and-dust-storm-hits-asia/, last access: 25 February 2024). During both dust events, dust originated from the central Gobi Desert and severely degraded the air quality across most parts of Mongolia and China. In the context of regional analysis of updated dust emission scheme, a case study is conducted. Figure 7 displays the daily mean DAOD from model results, MERRA-2 reanalysis products and MODIS derived results during the dust events. Both the original and updated schemes simulate the presence of large DAOD values in the Gobi Desert and the Taklamakan Desert on the 15th, 17th, and 28th, consistent with reanalysis and satellite data. According to the synoptic analysis, the impact area of dusting in the Taklamakan Desert is confined to the Tarim Basin due to the local easterly wind transport (Gui et al., 2022). The DAOD simulated by the original scheme notably exceeds that of the updated scheme, although the latter aligns more closely with the DAOD values of MERRA-2. The DAOD obtained from MODIS is consistently larger and suffers from discontinuity in this comparison.



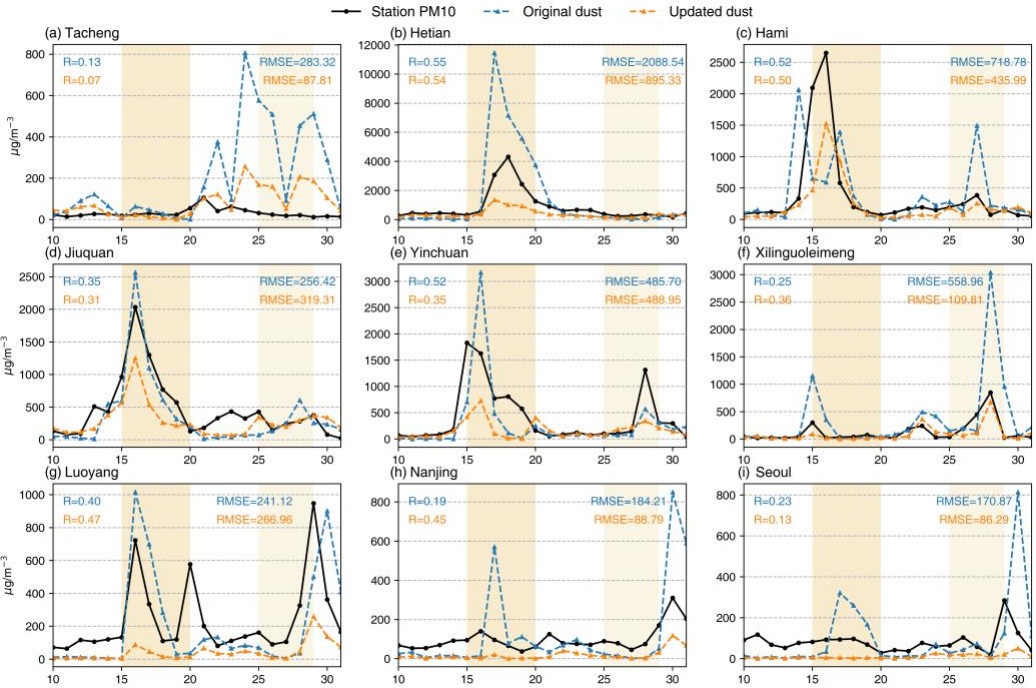

**Figure 8: Time series (a-i) of daily mean simulated dust aerosol concentrations (coloured lines) and station PM₁₀ concentrations (black lines) at selected stations (i). The blue and yellow lines represent the simulations from the original and updated schemes, respectively. The RMSE and R are noted in corresponding colored font for each simulation. The shading denotes the duration of the two dust events.**

The day-to-day evolutions of the PM$_{10}$ concentrations measured at the stations and the dust aerosol concentrations simulated by the model are displayed in Fig.8. Stations located near the dust source regions (Fig. 8a-f) experienced poor air quality during dust events, with PM$_{10}$ concentrations ranging from 1000 to 4000 μg m$^{-3}$. During these peaks in daily mean PM$_{10}$ concentrations, the updated scheme generally simulates smaller values compared to the original scheme, resulting in a closer match to observed values at Tacheng, Hetian, Hami, and Xilinguoleimeng stations. Notably, at the Hami station, while the original scheme incorrectly simulates two peak dust aerosol concentrations around March 15, the updated scheme aligns with observations by simulating one peak on March 16. Additionally, at Jiuquan and Yinchuan, stations situated at the southwestern edge of the Gobi Desert, both the original and updated schemes display false peak occurrences in simulations. Dust was transported downwind to middle and eastern China, such as Luoyang and Nanjing, which exhibits better simulation results with the updated scheme at both sites. This improvement comes from the updated scheme showing a weaker dust concentration at peak times. Observations at Seoul indicate that PM$_{10}$ concentrations reached 285 μg m$^{-3}$ on March 29, indicating long-range spread influence from the dust event. While the original scheme simulates peak dust aerosol concentrations during each of the two dust events at this site, the updated scheme only shows a minor increase on March 30. This discrepancy is attributed to the shorter dust aerosol residence time in the updated scheme, which limits the dust's transport over longer distances.





## 3.2 Sea-salt emission scheme

### 3.2.1 Model evaluation

| Sea-salt diameter (µm) | Original sea-salt emission (Tg yr⁻¹) | Updated sea-salt emission (Tg yr⁻¹) | Original sea-salt wet deposition[a] (Tg yr⁻¹) | Updated sea-salt wet deposition[a] (Tg yr⁻¹) |
|---|---|---|---|---|
| **0.02-0.08** | 0.6 | 0.02 | 0.3 (56%) | 0.009 (55%) |
| **0.08-1.0** | 97 | 63 | 77 (79%) | 51 (80%) |
| **1.0-10.0** | 2903 | 3122 | 2138 (73%) | 2310 (73%) |
| **0.02-10.0** | 3000 | 3185 | 2216 (73%) | 2361 (73%) |

**Table 2: Global sea-salt emissions and wet depositions in "CYCLE" simulation sets. a The ratio of wet deposition to total deposition (dry and wet deposition) is provided in parentheses next to wet deposition.**

With the modifications applied, we simulate a global annual total sea-salt emission of 3185 Tg, compared to 3000 Tg from original scheme simulation (Table 2). Results of the sea-salt emission are comparatively smaller than previous estimates from global models (Jaeglé et al., 2011; Liu et al., 2012; Spada et al., 2013; Weng et al., 2020), which suffer from significant

discrepancies due to different settings of sea-salt cut-off radius in aerosol schemes. Nevertheless, our results fall in the mid-range of values estimated from the historical CMIP5 simulations compiled in IPCC AR5 (1,400-6,800 Tg/year) (Intergovernmental Panel on Climate Change (IPCC), 2014). The simulated sea-salt burden is 9.1 Tg, slightly higher than 8.3 Tg using the original scheme. Both values are in agreement with the estimates by the AeroCom models (3.4 Tg-11.6 Tg) (Textor et al., 2006).

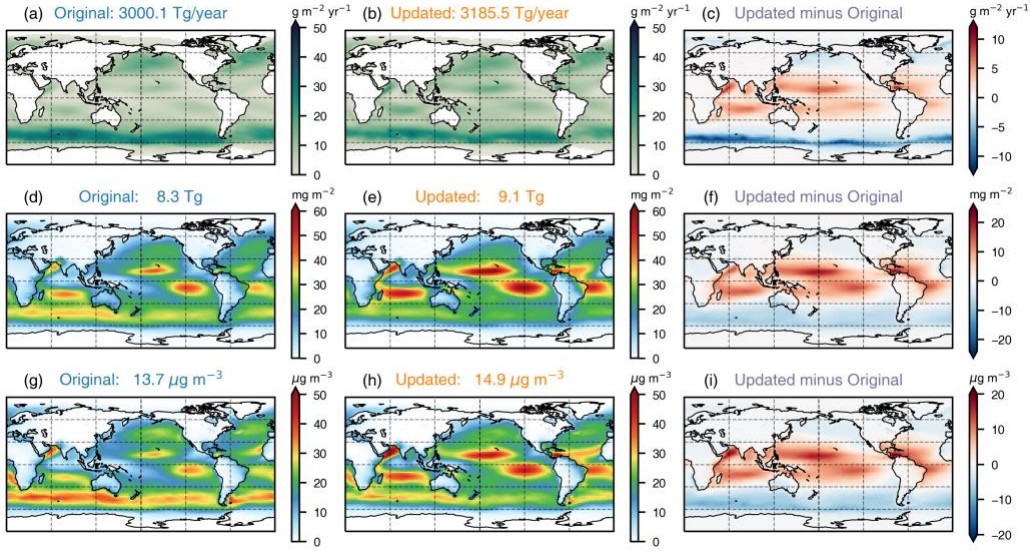


**Figure 9: Annual mean sea-salt emissions (a, b, c), burdens (d, e, f), and surface sea-salt aerosol concentrations (g, h, i) for the period from 2010 to 2012. Simulation results from the original (left) and updated (centre) emission schemes are shown, along with the differences between the updated and original schemes (right). The global total sea-salt emission, global total sea-salt burden, or global mean surface sea-salt concentration is also given at the top of each subplot.**



Figure 9 displays the global spatial distribution of annual sea-salt emissions, burdens and surface sea-salt aerosol
concentrations simulated with the default and modified schemes. The modifications to the sea-salt emission scheme exhibit a
clear impact on the emission distribution. In the Southern Ocean, where the original scheme simulates the most intense sea-
salt emissions, a notable decrease of 34 Tg yr$^{-1}$ (40°-65° S) is modeled by the updated scheme. Conversely, emissions increase
in tropical and subtropical oceans, particularly in the Arabian Sea within the Indian Ocean. These shifts in sea-salt emissions
correspond to changes in the spatial distribution of sea-salt burden and near-surface sea-salt aerosol concentration,
characterized by an increase in the region between 30° S and 35° N and a decrease outside this latitude range.

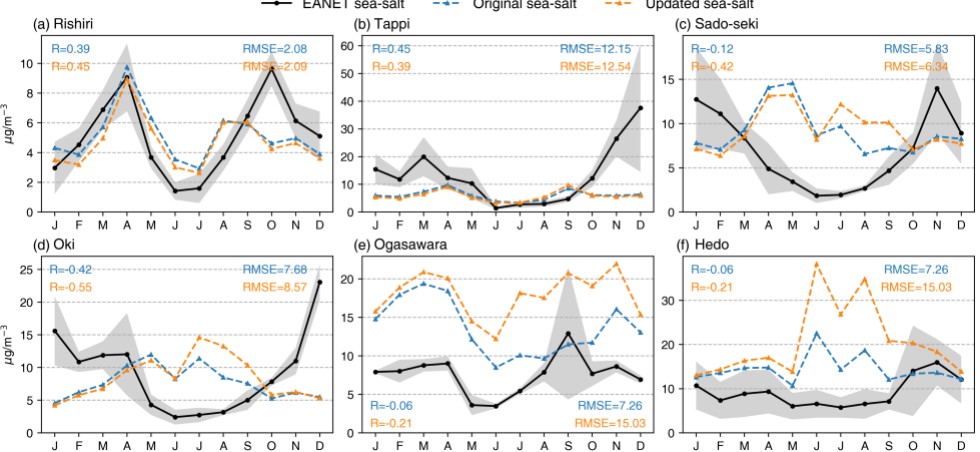

**Figure 10: Seasonal cycle of monthly mean simulated sea-salt aerosol concentrations (coloured lines) and EANET sea-salt concentrations (black lines) at selected stations. The locations of the stations are indicated in Fig.3. The blue and yellow lines**
**represent the simulations from the original and updated schemes, respectively. The RMSE and R are noted in corresponding colored font for each simulation. The shading on the observations illustrates the standard deviation of the monthly mean concentration over the months with sufficient data. Note that EANET sea-salt concentrations are calculated from Na$^+$ and Cl- ion concentration as (17) .**

We compare the simulation results of both schemes with the EANET aerosol concentration measurements (Fig. 10). Among
the listed stations (a-f, ordered by descending latitude), the original scheme captures the seasonal variations of sea-salt aerosol
concentrations at Rishiri, with an improvement in correlation observed in the updated scheme. Conversely, for the Tappi
station, there is little difference in results between the updated and original schemes, and neither reproduces the December
peaks in the observations. The updated scheme shows higher negative correlation coefficients in areas, such as Sado-seki and
Oki, where the results of the original scheme deviates significantly (with negative $R$) from the observations. We attribute the
bias at these two island stations to the low model resolution. Firstly, these islands remain unresolved by the 2-degree resolution
used. Secondly, the coarse resolution results in grid point values that are unrepresentative of the actual conditions at the
stations. Another island station, Ogasawara, situated furthest from the continent and least affected by continental anthropogenic
emissions, exhibits overestimation by both the original and updated schemes when compared with observed sea-salt aerosol
concentrations, although the June minimum is captured. However, the observed PM$_{10}$ concentrations at this station are well in
agreement with the modelled sea-salt aerosol concentrations of the original scheme (see Fig. S4e). In contrast, the updated





scheme simulation exhibits higher biases during months with higher SSTs (July to October, >25 °C), approximately 1.8 times. For the Hedo station on Okinawa Island, simulated sea-salt aerosol concentrations of the updated scheme differ from those of the original scheme by a factor of 1.8 on average during the months of high SST (June to September, >25°C). This is associated with the adoption of a different SST dependence function.

### 3.2.2 Dependence of the sea-salt emission scheme on SST and RH

| Sea-salt diameter (μm) | Original scheme (Tg yr$^{-1}$) | Gong (Tg yr$^{-1}$) | Gong + SST (Tg yr$^{-1}$) | Gong + RH (Tg yr$^{-1}$) |
|---|---|---|---|---|
| 0.02-0.08 | 0.6 | 0.03 | 0.02 | 0.02 |
| 0.08-1.0 | 97 | 76 | 64 | 76 |
| 1.0-10.0 | 2903 | 3761 | 3156 | 3773 |
| 0.02-10.0 | 3000 | 3837 | 3220 | 3849 |

**Table 3 Global sea-salt emissions in sensitivity simulations and a control run using the original scheme.**

We evaluate the impact of two major modifications on sea-salt aerosol emission schemes through a set of sensitivity experiments. Fig. 11 presents relative differences in annual mean emissions of submicron, coarse-mode, and total sea-salt aerosols between sensitivity simulations and the original scheme. The adoption of the extended *Gong* source function for sea-salt aerosol emissions, optimized for sea-salt particles smaller than 0.2 μm (Gong, 2003) leads to substantial changes in emissions of difference particle sizes. Specifically, the simulated emission of submicron sea-salt aerosol decreases, while that of coarse-mode sea-salt aerosol increases, as depicted in Fig.11b, d, g and the emissions listed in Table 3.

Firstly, we discuss the effect of the sea surface temperature (SST) correction factor on the modeled sea-salt emission. The simulation using the *Gong* function without the SST correction factor (Fig. 11a) predicts a reduction in submicron sea-salt aerosol emissions in low and middle latitude oceans, alongside an increase in high latitude oceans, compared to the original scheme. This contrasts with the simulation results of coarse-mode emissions using *Gong* function combined with SST correction proposed by Jaeglé et al. (2011) (Fig. 11e). This 'contrast' is in fact attributed to the fact that the original scheme employs a polynomial SST correction for sea-salt particles below 2.8 μm particle diameter and no SST constraints for particles with diameters larger than 2.8 μm. Thus, Fig. 11e demonstrates the effect of SST correction more straightforwardly. That is, in tropical oceans, sea-salt aerosol emissions are modeled to increase due to warm SST, while emissions are relatively suppressed in polar regions. The impact of SST constraint is negligible in mid-latitude oceans. Besides, upon applying the SST correction factor (Fig. 11b), the updated scheme simulates lower submicron sea-salt aerosol emissions globally, with major differences in the North Atlantic and regions affected by cold currents on the east coast of the Pacific Ocean. Yet, in the open ocean of the North Pacific, the tropical middle Pacific, and the tropical Indian Ocean, the *Gong* function combined with SST correction yields relatively minor differences from the original scheme.



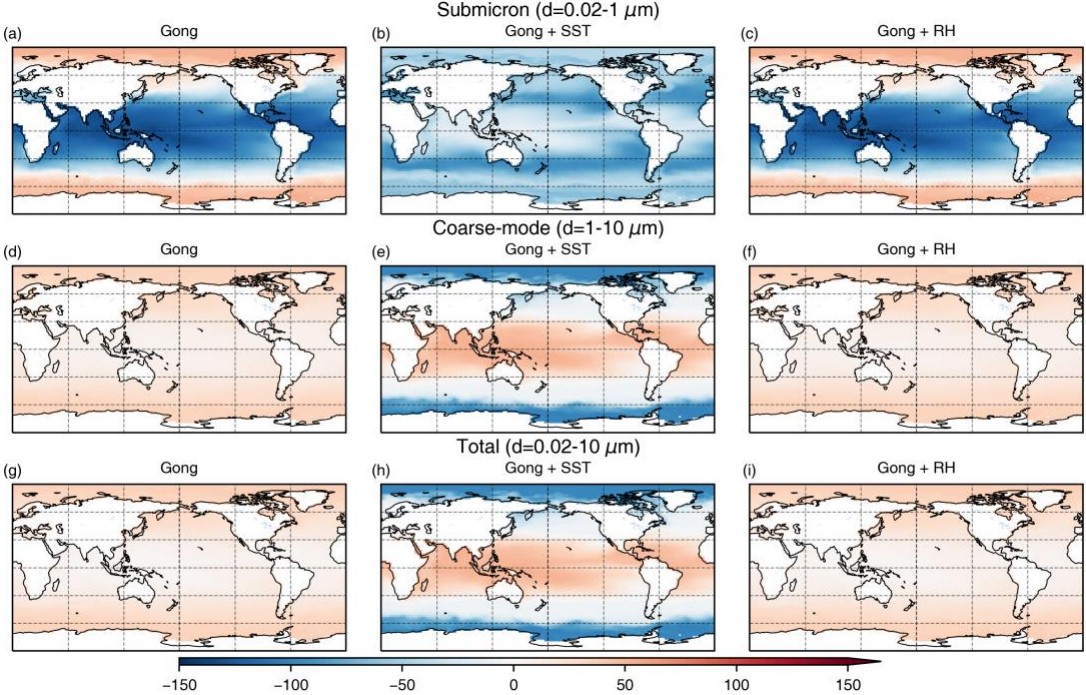

**Figure 11: Relative differences (%, per cent) in simulated annual sea-salt emissions between sensitivity simulations and the original scheme. The first row is for submicron sea-salt, the second for the coarse-mode sea-salt, and the third for the total sea-salt simulated. The aerosol size ranges are also given at the top of each panel.**

Turning to the effect of the relative humidity (RH) correction factor, comparison between the first and third columns in Fig. 11 reveals minimal impact on sea-salt emissions. Globally, there is a slight overall increase, averaging 0.3%. Still, the interplay between oceanic conditions and aerosol generation is intricate, with SST modulation showing a more pronounced impact than RH correction according to simulation results.

### 3.3 MPOA emission scheme

### 3.3.1 Model evaluation

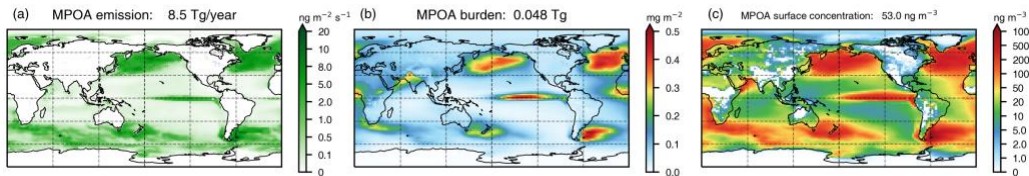

**Figure 12: Annual mean MPOA emissions (a), burdens (b), and surface MPOA concentrations (c) for the period from 2010 to 2012. The global total MPOA emission, global total MPOA burden, or global mean surface MPOA concentration is also given at the top of each subplot.**

The total mass of MPOA emitted globally is 8.5 Tg per year during 2010-2012 in our simulation. This result is within the

range estimated for submicron MPOA in previous modeling studies, from 2.3 Tg yr$^{-1}$ to 14.6 Tg yr$^{-1}$ (Burrows et al., 2018;

Gantt et al., 2011, 2012; Langmann et al., 2008; Meskhidze et al., 2011; Spracklen et al., 2008; Vignati et al., 2010). MPOA

burden is modelled as 0.048 Tg. Figure 12 shows the distribution of simulated annual MPOA emissions, burdens and surface

concentrations. Given the dependence on biological activity, the spatial pattern of MPOA emissions largely follows that of sea

surface chlorophyll concentrations (Fig. S5a). Maximum emissions are modelled to be in the eastern equatorial Pacific,

subtropical Pacific and Atlantic Oceans, and the Southern Ocean. The model result for the global mean sea surface [Chl $a$] is

0.16 mg m$^{-3}$, while the MODIS/Aqua satellite products suggest a global mean value of 0.45 mg m$^{-3}$. Note that the model output

for [Chl $a$] can influence the model performance in simulating MPOA emissions.

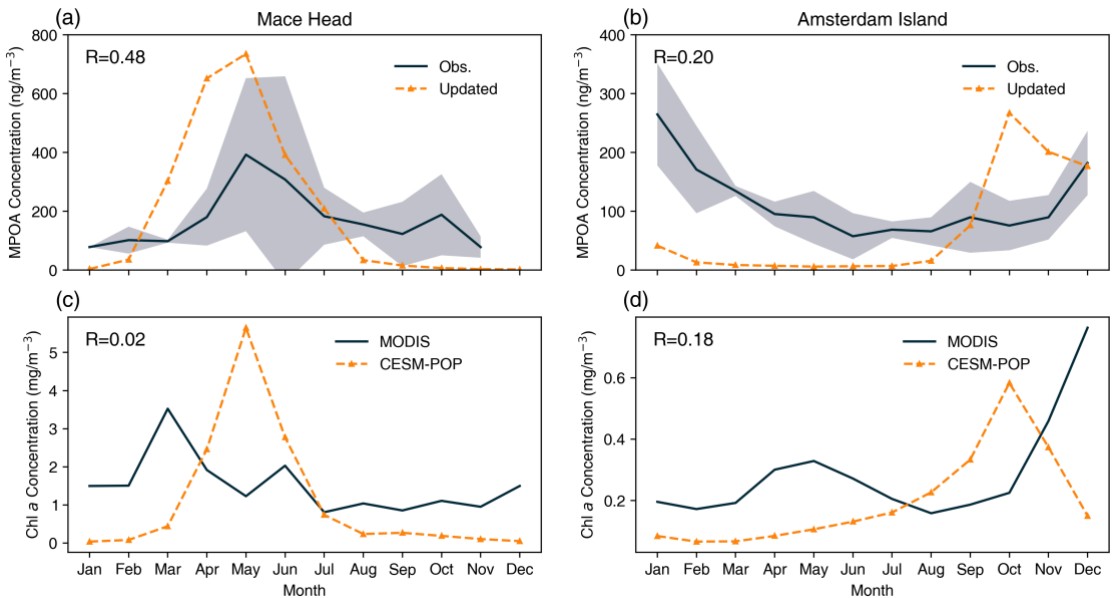

**Figure 13: Seasonal cycle of monthly mean simulated MPOA concentrations (coloured lines) and measured MPOA concentrations (black lines) at Mace Head (a) and Amsterdam Island (b). The locations of the stations are indicated in Fig.3. The Kendall's correlation coefficient (R) is noted. The shading on the observations illustrates the standard deviation of the monthly mean concentration over the months with sufficient data. Additional comparison of modeled and MODIS-derived [Chl $a$] at these locations is provided (c, d).**

We also evaluate the model simulation of MPOA concentrations using measurements from two representative sites. The first

site, Mace Head (53.33°N, 9.90°W), located near biologically productive waters in the North Atlantic Ocean, indicates high

MPOA concentrations from April to July in observations. However, the model simulates high concentrations from March to

July, capturing the peak in May but overestimating it by 1.9 times. Additionally, the model fails to reproduce other high

concentrations observed in October, possibly due to limitations in simulating [Chl $a$] (Fig. 13c). Another observation site is

Amsterdam Island (37.80°S, 77.57°E), which is subject to windy and biologically active currents in the Southern Ocean.

Observations shows a peak in January, whereas the model predicts that the peak occurs 3 months earlier, in October. Notably,





at both sites, the model underestimates MPOA concentration during months of low phytoplankton activity compared to measurements. By comparing the monthly mean [Chl *a*] from our POP2 simulation with MODIS-derived values at these locations (Fig. 13c, d), it can be seen that the modelled biases in MPOA corresponds to those in [Chl *a*] during months with lower [Chl *a*] levels. This suggests that the biases in MPOA simulations are closely tied to the biases in the modeled [Chl *a*], underlining the importance of efficient plankton representation in predicting MPOA concentrations. Another possible explanation could be the use of a uniform OM/OC value of 1.4. However, OM/OC values vary spatially and seasonally, typically ranging from 1.3 to 2.1 according to observations (Philip et al., 2014).

### 3.3.2 Effects of the phytoplankton species on MPOA emission

We illustrate the possible effects of phytoplankton species on MPOA emissions through a set of comparison experiments. Figure 14 shows the distribution of seasonal MPOA emissions modelled by [Chl *a*] from different phytoplankton species. Following the MARBL module that integrates marine biogeochemistry into the CESM2-POP2 component, phytoplankton is configured to represent in three functional groups: diatoms, diazotrophs, and small phytoplankton. These groups are distributed across global oceans based on factors such as nutrient limitation, light availability and temperature limitation as well as phytoplankton grazing or mortality. The resulting chlorophyll distribution is distinctive (refer to Fig. S7), shaping the modeled MPOA emissions in the atmospheric component model substantially.

In the boreal spring (MAM) and summer (JJA), MPOA emissions are simulated to peak in the North Pacific and North Atlantic regions, primarily due to the prevalence of diatoms and small phytoplankton, which are limited by iron and nitrogen nutrients in these areas. Conversely, during the boreal autumn (SON) and winter (DJF), elevated MPOA emissions are modeled in the oceans south of 30˚S, attributed to the growth of small phytoplankton in this region. Throughout the year, consistently high MPOA emissions are modeled in the eastern equatorial Pacific, driven by the high chlorophyll concentration in this region as simulated by POP2 models. However, satellite observations suggest that the chlorophyll concentration in this area is not as high as simulated (Fig. S5). Regarding diazotrophs, distribution of [Chl *a*] from this group is primarily concentrated in warmer sea areas as a result of temperature constraints. Due to the lower abundance compared to diatoms and small phytoplankton, diazotrophs have a less significant impact on MPOA emissions. The substantial role of biodiversity in shaping the composition of the Earth's atmosphere is reflected in our modelling results. However, the biological processes that produce these particles are poorly characterized, leading to large uncertainties in the estimation of global MPOA emissions.

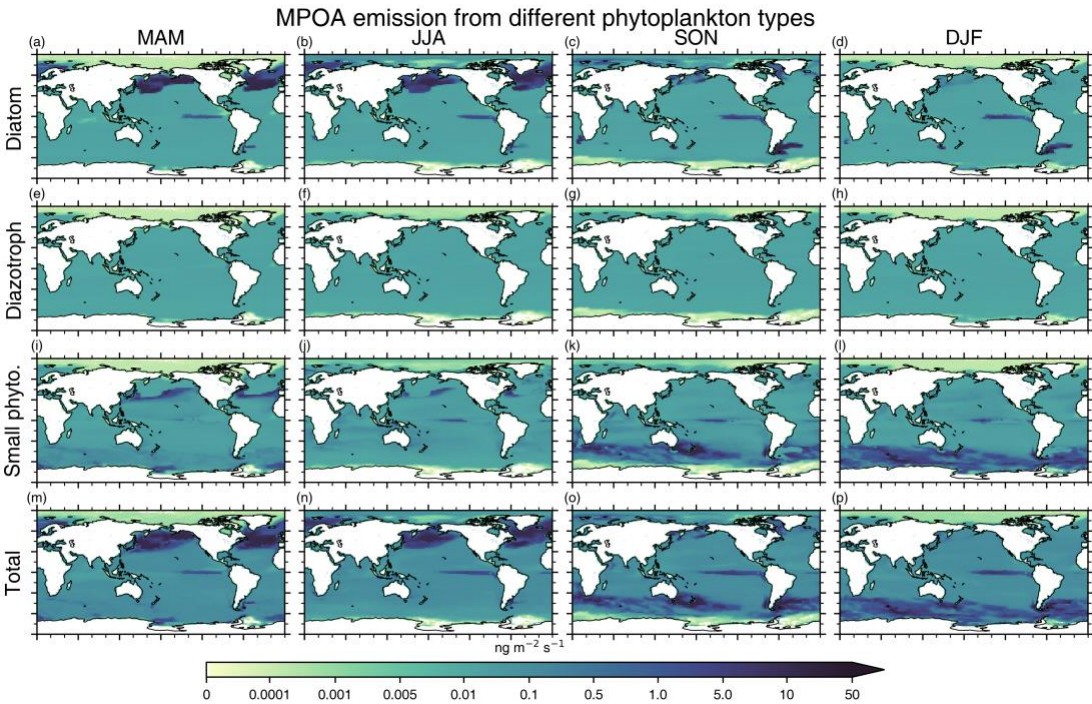

**Figure 14: Seasonal MPOA emissions resulting from [Chl *a*] of different types of phytoplankton. The first row is from diatom, the second from diazotroph, and the third from small phytoplankton (denoted as *Small phyto.*). The fourth row is the simulated emissions resulting from all the phytoplankton types above.**

### 3.4 Impact of dicarbonyls on SOA formation

The chemical pathways including gas-phase and heterogeneous reactions play a significant role in SOA formation. The impact of the dicarbonyls on SOA formation is illustrated by comparing the simulations with and without irreversible aqueous uptake of dicarbonyls in the chemical mechanism. Figure 15 shows the global distribution of surface SOA concentrations and the atmospheric burden simulated by the default (without the uptake) and modified (with the uptake) schemes. The Amazon, Central Africa, East Asia and Southeast Asia are the main regions with high surface concentrations of SOA in both schemes. The inclusion of the aqueous reaction pathway for SOA formation resulted in a global increase in surface SOA concentrations, with an average increase of about 37%. Growth is concentrated in Central Africa, East Asia and Southeast Asia (Fig. 15c), which is related to the spatial distribution of glyoxal and methylglyoxal (Fig. S8).



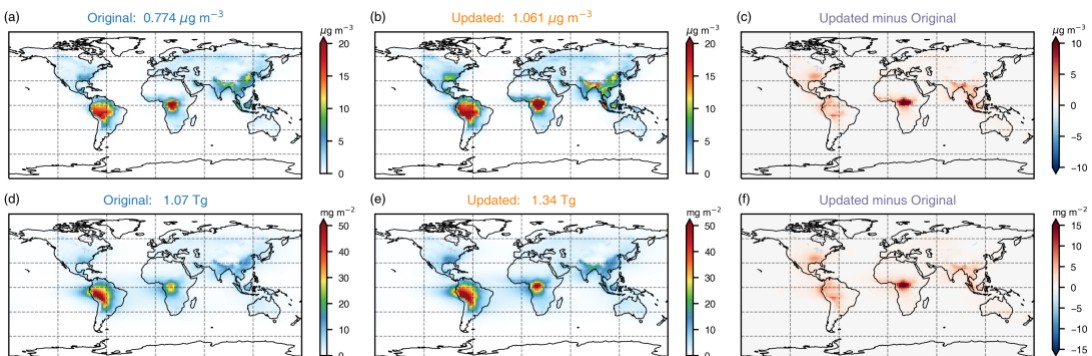

**Figure 15: Annual mean surface SOA concentrations (a, b, c) and burdens (d, e, f) for the period from 2010 to 2012. Simulation results from the original (left) and updated (centre) chemical schemes are shown, along with the differences between the updated and original schemes (right). The global mean surface SOA concentration, or global total SOA burden is also given at the top of each subplot.**


Differences in atmospheric burden, depositions, and lifetime of SOA between the two schemes are shown in Table 4. In the new scheme simulation, the total atmospheric burden and the depositions of SOA increased by about 25% and 30%, respectively. The lifetime of SOA against depositions is slightly reduced from 4.2 days to 4.1 days, which is consistent with Hodzic et al., (2016). The contributions of glyoxal and methylglyoxal to SOA formation are 14.3 Tg yr$^{-1}$ and 24.1 Tg yr$^{-1}$,

respectively, and occur mainly in the equatorial lower troposphere (Fig. S9). These values are higher than those estimated by Fu et al., (2008) in GEOS-Chem (6.4 Tg yr$^{-1}$ and 16 Tg yr$^{-1}$, respectively), which may be related to the differences in the simulation of dicarbonyl and aerosol surface area and other configurations (e.g. meteorology, emissions) between different models (Tsigaridis et al., 2014; Hodzic et al., 2020). Figure 16 shows the simulated global monthly surface mean concentrations of SOA and PM2.5 during 2010-2012. The model suggests that the irreversible aqueous uptake rate of dicarbonyls increases

notably (solid black line) when heavy haze events occur, resulting in a strong increase in SOA concentrations. The results indicate that the aqueous pathway through dicarbonyls can improve the underestimation of observed SOA concentrations during severe haze episodes (Huang et al., 2014; Li et al., 2021).

|  | Original scheme | Updated scheme |
|---|---|---|
| **SOA burden (Tg)** | 1.1 | 1.3 |
| **SOA dry deposition (Tg yr$^{-1}$)** | 13 | 17 |
| **SOA wet deposition (Tg yr$^{-1}$)** | 80 | 103 |
| **SOA lifetime (days)** | 4.2 | 4.1 |

**Table 4: Global atmospheric burden and depositions of SOA with (updated scheme) and without (original scheme) irreversible aqueous uptake of dicarbonyls in CAM6-Chem.**





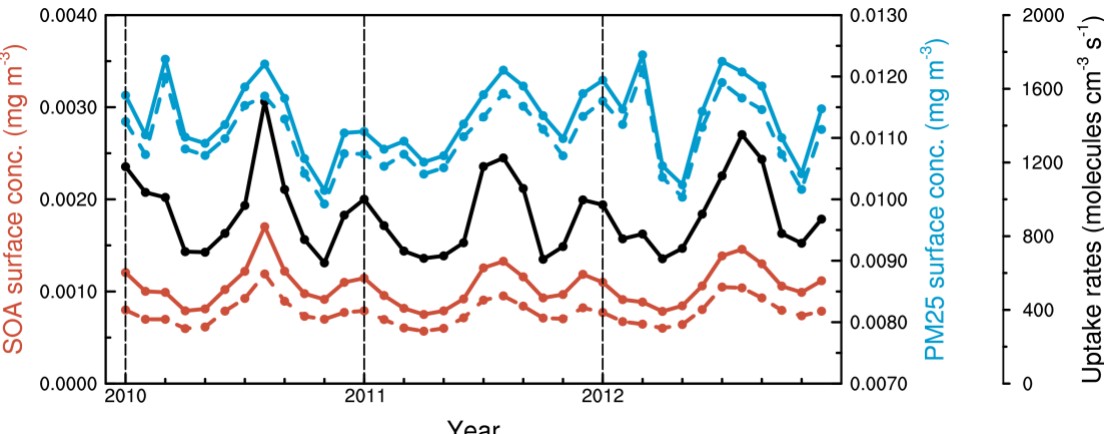


**Figure 16: The global monthly mean of SOA surface concentrations (red lines), PM2.5 surface concentrations (blue lines), and the uptake rates of dicarbonyls (black solid line) simulated in CAM6-Chem for the period from 2010 to 2012. The solid and dashed lines represent the updated and original schemes, respectively.**

## 4 Summary and conclusion

This study sets out to develop updated emission schemes for natural aerosol species based on the CoAerM, including dust, sea-salt, and MPOA, and SOA formation, including an irreversible aqueous uptake of dicarbonyls, in the CESM2. For dust emissions, the modified scheme confines dust deflation to erodible areas based on land use distribution instead of the original geomorphology-based hotspot-like source function, and integrates reduction factors for vegetation effects. Roughness length and soil texture from the land component, CLM5, is also incorporated to update threshold friction velocity correction factors.

The updated scheme yields a more continuous distribution of dust emission areas, and complement the emissions in North America and the sub-Arctic. Notably, DAOD simulations at stations in Central Asia (Karachi) and North Africa (Tamanrasset_INM) show more consistent alignment with observations in the updated scheme. Also, the updated scheme acts to shorten the residence time of dust aerosols, resulting in notable changes in simulated dust burden and associated DAOD simulations, particularly in downwind areas of the dust source region. The simulation of dust aerosol concentrations during

dust events is improved by the updated scheme in the downwind region of dust propagation. The sea-salt emission scheme is modified through updating the dependence of source function on SST and introducing a relative-humidity-dependent correction factor for sea-salt particle size. These modifications align emissions more intuitively with oceanic conditions and sea-salt production mechanisms. The modulation of sea-salt emission by SST is more pronounced in the simulations of the updated scheme, resulting in an increase in sea-salt emission in the tropical and subtropical oceans and a decrease in the

Southern Ocean. The RH correction factor exerts an enhancing effect across the globe, but the effect is very mild.

Moreover, we extend CESM's capabilities to capture the link between marine biology and atmospheric chemistry by including the MPOA emission scheme. Coupled offline with ocean component POP2, the representation of phytoplankton chlorophyll distribution by the ocean biogeochemistry module, MARBL, plays a crucial part in modelling MPOA emissions. Our



simulations reproduce the seasonal cycle observed at the North Atlantic station (Mace Head). However, the bias in the
simulation of the peak month at the Southern Ocean station (Amsterdam Island) may be related to the model's simulation of
the dominance of small phytoplankton in this region. We further compare the spatial variability of different phytoplankton
species on MPOA emission simulations, highlighting the significance of biological diversity in shaping aerosol emissions. For
the formation of SOA, we consider the irreversible aqueous uptake of dicarbonyl compounds (glyoxal and methylglyoxal) in
the chemical mechanism. The results show that this pathway makes an important contribution to the surface SOA
concentrations, especially during severe haze events. The accurate simulation of SOA needs further research into incorporating
additional processes and optimizing model parameters. Collectively, these modifications make the CESM a comprehensive
tool for elucidating the complexities of aerosol emissions and transformation from different spheres in the Earth system, such
as the land and ocean, thus facilitating the potential for improved evaluation of their impacts on climate processes and
feedbacks.


**Code availability**

CESM2 is an open-source community project and the model codes of version 2.1.3 are available at
https://github.com/ESCOMP/CESM/tree/release-cesm2.1.3 (last access: 5 June 2024). The code used for the updated aerosol
schemes in this study is permanently archived on Zenodo at https://doi.org/10.5281/zenodo.11488849 (last access: 5 June
655  2024).

**Data availability**

Output data from the model are available from the authors upon request. The data used in this article are available as follows:

- Daily MODIS/Aqua Level 3 product (https://ladsweb.modaps.eosdis.nasa.gov/archive/allData/61/MYD08_D3/, last
  access: 5 June 2024)
- DAOD climatology derived from MODIS retrievals
  (https://drive.google.com/drive/folders/1r7H_h_hiynx89vncOmGrozKFjTRTE4BR, last access: 5 June 2024)
- AERONET measurements of AOD (https://aeronet.gsfc.nasa.gov/cgi-bin/webtool_aod_v3, last access: 5 June 2024)
- MERRA-2 reanalysis of DAOD (https://goldsmr4.gesdisc.eosdis.nasa.gov/data/MERRA2/M2T1NXAER.5.12.4/,
  last access: 5 June 2024)



## Author contributions

YW implemented updates of emission schemes of dust, sea-salt, and MPOA, ran the simulation, performed the model evaluation, investigation, and formal analysis. PZ implemented the updated SOA formation scheme and performed the analysis. YZ, JWL and ZH conceived of the project and acquired funding. JL and YL provided the observational data of SOA and commented on the related discussion. JWL contributed EANET data. YW performed the visualization and prepared the original draft with input from PZ. Review and editing were performed by all co-authors.

## Competing interests

The authors declare that they have no competing interests.

## Acknowledgement

This study has been supported by the National Key Research and Development Program of China (grant no. 2019YFA0606803). The authors thank all the scientists, software engineers, and administrators who contributed to the development of CESM project. We also appreciate the science teams of EANET, AERONET, MODIS, and VIIRS for their work related to data maintenance.

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
