# Peer review of "An updated aerosol simulation in the Community Earth System Model (v2.1.3): dust and marine aerosol emissions and secondary organic aerosol formation"

_Geoscientific Model Development, 2024_

## Author Comment (AC1)

**Response to Reviewer 1:**

We would like to thank Reviewer 1 for dedicating time to carefully read our manuscript and provide feedback. We sincerely think their detailed comments have helped us to improve the manuscript. Here it follows a point-by-point response to the reviewer's report (text in black denotes the comments provided, while text in blue denotes our response), associating with the revised manuscript with the track of the changes.

**Comments (in black):** This study attempted to improve the aerosol simulation in the CESM model by updating the emission schemes for dust and marine aerosols as well as the heterogeneous chemistry of SOA formation. These updates were further supported by evaluating model results against multiple aerosols measurements. Overall, this is an important study and the overall conclusions are reasonable. I would like to recommend its publication after further revisions. In particular, the presentation quality of this manuscript should be improved. Please find my comments in the following:

**Response (in blue):** We sincerely appreciate Reviewer 1 for the thoughtful suggestions and constructive feedback, which have greatly contributed to the improvement of our paper. In the following responses, we have addressed all the comments and made revisions accordingly, which we believe have led to a significantly improved manuscript.

In Section 2.2, it is strongly suggested to add a Table to list all the model simulations conducted in this study.

Thank you for your constructive suggestion. A new table was added around L283, listing all simulations discussed in the manuscript and their related information, as follows.

**Table 1: List of all simulation experiments in this study.**

| Experiment set | Annotation/Name | Horizontal resolution | Brief descriptions |
|---|---|---|---|
| **CYCLE** | CYCLE-original | $1.9° \times 2.5°$ | 2009-2012, CAM6-chem default scheme |
| | CYCLE-updated | $1.9° \times 2.5°$ | 2009-2012, updated scheme in this study |
| Case study of dust events | Case-original | $0.9° \times 1.25°$ | 1 January 2021 to 1 April 2021, CAM6-chem default dust emission scheme |
| | Case-updated | $0.9° \times 1.25°$ | 1 January 2021 to 1 April 2021, updated dust emission scheme |
| Sensitivity experiments on sea-salt aerosol scheme | SS-Gong | $1.9° \times 2.5°$ | 2009-2012, *Gong* source function |
| | SS-Gong+SST | $1.9° \times 2.5°$ | 2009-2012, *Gong* function together with SST-dependent correction factor |
| | SS-Gong+RH | $1.9° \times 2.5°$ | 2009-2012, *Gong* function together with RH-dependent correction factor |

| | MPOA-diatom | 1.9° × 2.5° | 2009-2012, [Chl *a*] input only from diatom |
|---|---|---|---|
| Sensitivity experiments on MPOA scheme | MPOA-diazotroph | 1.9° × 2.5° | 2009-2012, [Chl *a*] input only from diazotroph |
| | MPOA-small phyto. | 1.9° × 2.5° | 2009-2012, [Chl *a*] input only from small phytoplankton |

Fig.10 compared the simulated and observed sea-salt concentrations, but it looks the model performance is good. This model bias is mainly attributed to the coarse model resolution. Why is model resolution particularly critical for sea-salt simulation? In addition, the updated simulation shows improved correlation but monthly variations of sea-salt aerosols in the two simulations look like very similar. I suggest the authors to check over the calculated correlation.

We appreciate your comments regarding the model bias and the importance of resolution in sea-salt simulations.

1. We agree that model resolution is critical for accurately simulating sea-salt aerosols. The coarse resolution limits the model's ability to capture fine-scale variability in wind speed, which is crucial for simulating sea-salt aerosol generation. This can lead to biases in sea-salt concentration, especially in regions with complex coastal topography or variable oceanic conditions.
We rephrased the discussion to address these issues as follows:
L494-499 "*We attribute the bias at these two island stations to the low model resolution. Firstly, the 2-degree resolution used in the model is insufficient to resolve these islands, making it difficult to accurately represent the specific conditions at the stations. Secondly, the coarse resolution results in grid point values that do not accurately reflect the actual conditions, particularly affecting the model's ability to capture fine-scale variability in wind speed. This limitation is critical for simulating sea-salt aerosol generation, as fine-scale wind variations are essential in regions with complex coastal topography or variable oceanic conditions.*"

2. Regarding the similarity in monthly variations between the two simulations, we acknowledge that while the correlation improved, the overall pattern remained consistent between the original and updated schemes. This is likely because the primary source functions and meteorological inputs, which drive the monthly variability, are similar in both simulations. We have re-examined the calculated correlation (R) to ensure the accuracy.

As shown in Fig.15, the effects of added SOA formation pathways are notable over biogenic and biomass burning-affected region. As such, I suggest to show some seasonal variations of these effects.

Thank you for your insightful suggestion. We added some descriptions in L609-614 and a figure in SI (Figure S9) to show the seasonal variations of these effects.
"*In addition, we find that the effect of dicarbonyls on SOA formation shows significant seasonal*

*variation, with higher contributions in boreal summer (JJA) and winter (DJF) and relatively lower contributions in spring (MAM) and autumn (SON). Regionally, high values in summer are mainly observed in Southeast Asia, North America, and the Amazon, while in winter, they are concentrated in Central Africa and South Asia (Fig. S9a). Biogenic emissions of isoprene, the primary precursor of dicarbonyl compounds (Fu et al., 2008; Kelly et al., 2018), are the main drivers of these spatiotemporal variations (Fig. S9b)."*

In Abstract and Conclusions Sections, I strongly suggest the authors to add some quantitative conclusions that show the effects of update schemes on aerosol simulations.

Thank you for your constructive suggestion. In the revised manuscript, we incorporated specific numerical results to provide a clearer understanding of their significance.

[revised manuscript text omitted]

The table caption should be moved to the top of each table.

Corrected throughout the manuscript.

I think the "CAM6-Chem" should be changed to "CAM6-chem" throughout the text.

We agree and have thoroughly re-examined the use of the terms throughout the revised manuscript.

L263: The reference for MERRA2 reanalysis data looks not correct.

We reviewed the citation requirements for this dataset, and the corrections are as follows: (NCAR/UCAR, 2018)
Atmospheric Chemistry Observations Modeling/National Center For Atmospheric Research/University Corporation For Atmospheric Research and Climate And Global Dynamics Division/National Center For Atmospheric Research/University Corporation For Atmospheric Research: MERRA2 Global Atmosphere Forcing Data [data set], https://doi.org/10.5065/XVAQ-2X07, 2018.

L267: The "CYCLE" simulation was using CAM5 not CAM6?

We apologize for the misunderstanding caused by the typo here. The correct term is CAM6-chem, and this has now been updated in the text.

L302-303: I suggest to move the introduction of model evaluation metrics after Section 2.3.

We reorganized Section 2.3 to place the introduction of model evaluation metrics at the end of this section:
L327-329 "*In the following discussion, the evaluation metrics used are the Kendall's correlation coefficient (R) and root mean square error (RMSE). Kendall's correlation, which does not assume a specific data distribution, is used to assess the statistical dependence between observed and simulated values. RMSE measures the average error between observation and simulated results.*"

L316: Why were these two measurement sites considered here?

Thank you for your question regarding the selection of the two measurement sites. Mace Head and Amsterdam Island were chosen due to their geographical locations and the availability of long-term observational data. Mace Head represents a Northern Hemisphere mid-latitude site, while Amsterdam Island provides a contrasting Southern Hemisphere marine environment. These sites offer valuable insights into the performance of our model across different hemispheres and oceanic conditions. Additionally, these sites are commonly used in previous studies for comparing MPOA simulations, making them suitable for consistent and meaningful evaluation of our model.
We clarified this rationale in the revised manuscript as follows:
Starting from L564 "*We also evaluate the model simulation of MPOA concentrations using measurements from two representative sites. The first site, Mace Head (53.33˚N, 9.90˚W), located near biologically productive waters in the North Atlantic Ocean……Another observation site is Amsterdam Island (37.80˚S, 77.57˚E), which is subject to windy and biologically active currents in the Southern Ocean…….*"

L461: Any updated model results from CMIP6?

Thank you for pointing this out. We agree that referencing the most recent CMIP6 data would provide additional context and situate our findings within the latest climate modeling framework. We included a comparison with the updated CMIP6 model results in the revised manuscript as follows:

L463-466 *"Nevertheless, our results fall in the mid-range of values estimated from the historical CMIP5 simulations compiled in IPCC AR5 (1400-6800 Tg/year) (Intergovernmental Panel on Climate Change (IPCC), 2014), and are also consistent with the broader range observed in CMIP6 simulations (2624–64939 Tg/year) (Thornhill et al., 2021)."*

L610: The authors only show the changes in global annual mean PM2.5 but attempt to link it with haze episodes. Any further evidence to support this argument?

We clarified this by modifying the sentence in L630-636 as:

*"Previous studies have shown that during heavy haze episodes, organic aerosols can account for up to half of the PM mass, with a significant contribution from SOA (Huang et al., 2014, Zhao et al., 2019). Figure 16 shows the simulated global monthly surface mean concentrations of SOA and PM2.5 during 2010-2012. The model suggests that the irreversible aqueous uptake rate of dicarbonyls increases notably (solid black line) when heavy haze events occur, resulting in a strong increase in SOA concentrations. The results indicate that the aqueous pathway through dicarbonyls can improve the underestimation of observed SOA concentrations during severe haze episodes (Li et al., 2019; Li et al., 2021)."*

In Fig.6, I suggest to add the location information (e.g., Latitude and longitude) of each AERONET site on the plots.

Thank you for your suggestion. We incorporated this information in the revised manuscript to enhance the clarity and utility of the figure.

[Figure]

In Fig.13, to be consistent with other Figures, I suggest to also add the RMSE metric in the plot.

Thank you for your suggestion. We added the RMSE metric in Fig. 13 (also the site location information) to enhance consistency across the figures and allow for easier interpretation of the results.

[Figure]

**Additional changes:**

- Minor wording adjustments and corrections throughout the manuscript.
- Added annotation to the right side of each panel in Figure 4, 9, and 15 to provide clearer context.
- Standardized the formatting of PM2.5 in the text to display as $PM_{2.5}$ in subscript.

We hope that we have adequately addressed all the suggestions raised by Reviewer 1, and appreciate their constructive feedback

---

## Author Comment (AC2)

**Response to Reviewer 2:**

We would like to express our gratitude to Reviewer 2 for their thorough review of our manuscript and their attention to detail in the language used, particularly considering the manuscript's length. We carefully went through the whole manuscript again and to our best knowledge have identified and corrected all typos and mistakes therein. Below, we outline in detail how we have addressed the comments provided (text in black denotes the comments provided, while text in blue denotes our response):

**Comments (in black):** This manuscript addresses the improvement of aerosol simulations in the CESM model by revising emission schemes for dust and marine aerosols and incorporating aqueous chemistry for SOA formation. The authors present a clear and systematic approach, starting with the implementation of revised aerosol schemes, followed by the comparison of their updated simulation results with various aerosol observational measurements to evaluate their proposed simulations. In particular, the authors design sensitivity experiments in a further discussion to capture the uncertainties in the simulations of these several aerosol species. Given the focus of the study, the methodologies used, and the conclusions presented, this manuscript meets the criteria for publication in Geoscientific Model Development.

However, the manuscript in its current form requires better organization and clarity. I recommend publication after the following revisions are made:

**Response (in blue):** We appreciate Reviewer 2 for the useful suggestions that have helped us to improve our paper. As indicated in the responses that follow, we have taken all these comments and suggestions into account in the process of revision.

Page 2 Line 33: change "indicate" to "indicates".

Corrected.

Page 2 Line 43: change "is" to "are".

Corrected.

Page 2 Line 52: add comma before "and even…".

Corrected.

Page 3 Line 87: change "comprising of" to "comprising".

Corrected.

Page 3 Line 90: add "the" before "Earth system". Change "is" to "are".

Corrected.

Page 4 Line 114: remove "the" before "Owen's effect".

Corrected.

Page 6 Line 143: remove "By" before "corresponding to".

Corrected.

Page 6 Line 154: change "shows" to "show".

Corrected.

Page 7 Line 173: change "region" to "regions".

Corrected.

Page 8 Line 197: change "varies" to "vary".

Corrected.

Page 9 Line 226: add "the" before "majority".

Corrected.

Page 10 Line 249: change "dicarbonyls" to "dicarbonyl".

Corrected.

Page 10-11 Section 2.2: The authors mentioned that a "CYCLE" experiment, along with a case study and several other sensitivity experiments, was conducted. To enhance readability, a table-like presentation should be included.

Thank you for your constructive suggestion. We agree that to present the "CYCLE" experiment, along with the case study and other sensitivity experiments, in a table format would improve the readability and clarity of this section. We included a Table around L283, summarizing the key details of these experiments, as follows.

**Table 1: List of all simulation experiments in this study.**

| Experiment set | Annotation/Name | Horizontal resolution | Brief descriptions |
|---|---|---|---|
| **CYCLE** | CYCLE-original | $1.9° \times 2.5°$ | 2009-2012, CAM6-chem default scheme |
| | CYCLE-updated | $1.9° \times 2.5°$ | 2009-2012, updated scheme in this study |
| Case study of dust events | Case-original | $0.9° \times 1.25°$ | 1 January 2021 to 1 April 2021, CAM6-chem default dust emission scheme |

| | Case-updated | 0.9° × 1.25° | 1 January 2021 to 1 April 2021, updated dust emission scheme |
|---|---|---|---|
| Sensitivity experiments on sea-salt aerosol scheme | SS-Gong | 1.9° × 2.5° | 2009-2012, *Gong* source function |
| | SS-Gong+SST | 1.9° × 2.5° | 2009-2012, *Gong* function together with SST-dependent correction factor |
| | SS-Gong+RH | 1.9° × 2.5° | 2009-2012, *Gong* function together with RH-dependent correction factor |
| Sensitivity experiments on MPOA scheme | MPOA-diatom | 1.9° × 2.5° | 2009-2012, [Chl *a*] input only from diatom |
| | MPOA-diazotroph | 1.9° × 2.5° | 2009-2012, [Chl *a*] input only from diazotroph |
| | MPOA-small phyto. | 1.9° × 2.5° | 2009-2012, [Chl *a*] input only from small phytoplankton |

Page 11 Line 271: remove the article "a" before "dust events".

Corrected.

Page 11 Line 278: change "a" to "an".

Corrected.

Page 11 Line 280: change "involves" to "involve".

Corrected.

Page 12 Line 302: Two statistical metrics are mentioned here for model evaluation. A description of these two metrics should be added, and their use should be consistent in other similar time series comparisons throughout the manuscript.

Thank you for your suggestion. We added a description of the two statistical metrics mentioned to clarify their definitions and significance in the context of our model evaluation:
L327-329 "*In the following discussion, the evaluation metrics used are the Kendall's correlation coefficient (R) and root mean square error (RMSE). Kendall's correlation, which does not assume a specific data distribution, is used to assess the statistical dependence between observed and simulated values. RMSE measures the average error between observation and simulated results.*"

We also revised Figure 13 to make sure these metrics are consistently applied in all similar time series comparisons.

Page 17 Figure 7: The color bar needs adjustment. The current maximum value doesn't adequately represent the information in certain areas.

Thank you for pointing out the issue with the color bar in Figure 7. We adjusted the maximum value to ensure that the range of color bar more accurately represents the information in the relevant areas.

[Figure]

Page 22 Line 524: add "a" before "comparison".

Corrected.

Page 24 Line 558: change "corresponds" to "correspond".

Corrected.

**Additional changes:**
- Minor wording adjustments and corrections throughout the manuscript.
- Added annotation to the right side of each panel in Figure 4, 9, and 15 to provide clearer context.
- Standardized the formatting of PM2.5 in the text to display as $PM_{2.5}$ in subscript.

We hope that we have adequately addressed all the suggestions raised by Reviewer 2, and appreciate their constructive feedback